# Identification of Droughts and Heat Waves in Germany with Regional Climate Networks

Gerd Schädler and Marcus Breil

Institute for Meteorology and Climate Research - Department Troposphere Research, Karlsruhe Institute of Technology, Karlsruhe, Germany

**Correspondence:** Gerd Schädler (gerd.schaedler@kit.edu)

**Abstract.**

Regional Climate Networks (RCNs) are used to identify heat waves and droughts in Germany and two subregions for the summer half years and summer seasons of the period 1951 to 2019. RCNs provide information for whole areas (in contrast to the point-wise information from standard indices), the underlying nodes can be distributed arbitrarily, they are easy to construct and provide details otherwise difficult to avail of like temporal and spatial extent and localisation of extreme events; this makes them suitable for the statistical analysis of climate model output. The RCNs were constructed on the regular 0.25 degree grid of the E-Obs data set. The season-wise correlation of time series of daily maximum temperature $T_{max}$ and precipitation were used to construct the adjacency matrix of the networks. Based on the results of a sensitivity study, we used as main metrics to characterise the network structure the edge density, which increases significantly during extreme events. The standard indices for comparison were the effective drought and heat index (EDI and EHI) respectively, based on the same time series, and complemented by other published data. Our results show that the RCNs are generally able to identify severe and moderate extremes and can differentiate between regions as well as seasons.

## 1 Introduction

Extreme events such as heat waves, droughts and floods are causing casualities, severe damage and economic losses. It is predicted that the frequency, duration and intensity of such extremes will increase during this century in several European regions, already affected ones such as in the Mediterranean as well as new ones in midlatitudes (Beniston et al., 2007). Knowledge about the present state and future changes of extremes is of great importance both from the scientific (process understanding) as well as from a societal standpoint (adaptation and mitigation measures) perspective. It would therefore be very useful to have a fast and easy-to-apply tool to identify extremes, vulnerable regions and critical seasons.

To identify extreme events, several extreme indices have been developed, like the Standardised Precipitation Index (SPI) for floods, the Universal Thermal Climate Index (UTCI) for heat waves and the Palmer Drought Index (PDI) and the Effective Drought Index (EDI) for droughts, see for instance the WMO guideline for precipitation and temperature extremes (ETCCDI, http://www.wmo.int/pages/prog/wcp/wcdmp/CA_3.php). These indices are used to produce catalogues of extreme events like the ones published by the European Drought Observatory (https://edo.jrc.ec.europa.eu/edov2/php/index.php?id=1000). How-

ever, these indices differ considerably in purpose, timescales of interest, methods used, thresholds and therefore also in events considered extreme (Byun and Wilhite, 1999).

We propose here a method to identify extremes based on regional climate networks (RCNs) which can be applied to various types of extremes, is easy to apply and computationally efficient, has very few tuning parameters and permits a fast analysis for whole regions instead of single points (as do most commonly used indices). Beyond complementing existing methods, it offers several advantages. Applying the method to the results of regional climate models, present day and future vulnerable regions can be detected. By using community detection methods and studying the temporal dynamics of the network structure (not done in this paper), the importance of processes affecting the occurence of extremes like weather patterns, continentality, orography and land use can be assessed. The attribute "regional" means that the nodes of these networks are confined to a geographical region as opposed to the whole globe, similar to the difference between regional and global climate models; it is indeed our ultimate goal to apply the RCNs to the output of regional climate models to produce statistics of extreme events/episodes. We are interested here in extreme events happening in regions larger than a minimum size, i.e. in the order of $10^4$ to $10^5$ $km^2$ and are coherent and collective, i.e. most sites in such a region are affected in a similar way, so that the time series (extended over several months) of the relevant variables (daily maximum temperature, dry days) are highly correlated during extreme events; therefore, correlation coefficients above a given fixed threshold will be used to construct the RCNs in this study.

The general idea, then, of climate networks is to consider geographical points, which can be the grid points of reanalysis data, of a climate model, or a network of observation sites, as nodes of the network. A link between two nodes exists if the statistical association measure (e.g. the Pearson correlation coefficient) between the time series of the variables exceeds a given threshold. From this, one obtains the so-called adjacency matrix, which is essentially a list of connected nodes. Metrics of this adjacency matrix like node degree, edge density and clustering coefficient can then be used as indicators for extreme events like heat waves, floods and droughts (see e.g. Tsonis et al. (2006)).

The study of networks has evolved from graph theory; so-called random networks have been studied mathematically by Erdös and Renyi (Erdös and Rényi (1959)). Soon they were recognised as a very useful tool to analyse real-world networks like electricity grids or the internet and assess their vulnerability. An overview over networks in general and their various applications in different disciplines can be found e.g. in Newman (2003), Newman (2019), Watts and Strogatz (1998) and Albert and Barabási (2002).

Climate networks have been increasingly used in recent years, initially mainly in a global context. They were applied to study global oscillation patterns like El Niño and to reveal teleconnections by Donges et al. (2009) (this paper also contains definitions of higher-level network metrics). Tsonis and Swanson (2012) used climate networks to study decadal climate variability, Ludescher et al. (2013) developed a network method to improve El Niño forecasting, and Boers et al. (2014) did so for the prediction of extreme floods. It has also been shown that climate networks are able to extract interesting information about climate processes, e.g. the relation between climate and topography (Peron et al., 2014). Overviews over the application of networks to climate can be found in Dijkstra et al. (2019) and the review by Donner et al. in Franzke and O'Kane (2017). There is also an increasing number of applications of climate networks to regional scales. Rheinwalt et al. (2016) studied the spatial synchronisation of precipitation in Germany using a regional climate network. They calculated precipitation isochrones

and could identify fronts along which heavy precipitation events propagated. In a similar vein, Mondal and Mishra (2021) used a regional network to analyse and predict heat wave clusters and the propagation of heat wave fronts over the United States. Weimer et al. (2016) used a regional climate network to predict future heat periods in Europe on decadal time scales; they found that the network approach is in some regions and decades superior to the standard approach to estimate the occurence of

heat periods. More applications (regional, oceanic and atmospheric studies) can be found in the overviews mentioned above.

In the present study, we use RCNs to analyse the occurence of past heat and drought extremes in Germany and show that they have the potential to describe the occurence frequency and spatial extent of droughts and heat waves. Our working hypothesis is that extremes like heat waves and droughts are characterised by spatial and temporal coherence which is reflected in the metrics of suitably constructed regional climate networks. We will focus mainly on the edge density as the most immediate

metric, which we expect to peak for seasons where extremes occur.

If one has such a tool, it can be integrated routinely and efficiently into the postprocessing of climate simulations to establish climatologies of extremes (specifically heat waves and droughts) on regional scales for a given season at yearly or decadal resolution; it could also be used to routinely analyse regional climate model results (especially climate prognoses) to identify vulnerable regions, seasons prone to extremes and trends in extremes, for example. From the process understanding perspective,

studying the structure of the adjacency matrix permits assessing "noise" factors like orography, land use, continentality and weather patterns.

This study should be considered a proof-of-concept study; we will study the sensitivity of the RCN to its construction and then apply the RCN to comparisons with present-day observations; our ultimate goal (not presented in this paper) is to apply RCNs to projections of regional climate models in various regions to assess future changes of extremes.

This paper is structured as follows: in section 2, we describe the construction of networks and introduce the metrics used. We also present the data and reference extreme catalogues used as well as the regions considered. In section 3, we study the sensitivity of the network to the choice of the correlation threshold. In section 4, we present comparisons of heat waves and drought extremes identified with RCNs with standard indices and discuss the effects of chosen regions and season. A summary is given in section 5.

## 2   Methods and data

### 2.1   Construction of RCNs and metrics used

We describe here only those aspects of climate networks which are relevant to our study; for more information on networks in general, the reader is referred to e.g. Newman (2003), Newman (2019), Watts and Strogatz (1998) or Albert and Barabási (2002); climate networks are described e.g. in Tsonis et al. (2006), Dijkstra et al. (2019), Donner et al. in Franzke and O'Kane

(2017) and Donges et al. (2009). For the definition of edges, i.e. the construction of the adjacency matrix, the pairwise statistical similarity of the nodes must be quantified. For this purpose, several measures are available; frequently used ones are the Pearson correlation coefficient, event synchronisation (Boers et al. (2014)) and mutual information (e.g. Donner et al. in Franzke and O'Kane (2017)). In this study, we used the Pearson correlation coefficient $\rho$. We construct our RCNs, i.e. adjacency matrices,

as undirected graphs with grid points of a regular lon-lat grid as nodes; two nodes are connected by an edge if the correlation $\rho$ of time series of the daily maximum temperature $T_{max}$ for heat waves and dry days for droughts, respectively between the two nodes exceeds a predefined threshold value $\rho_0$. The effect of the choice of $\rho_0$ will be discussed in section 3.

The structure of our RCN is thus determined by the strength of the correlation between the nodes which has to exceed the prescribed and fixed correlation threshold, so that all metrics can vary from year to year. This approach is different from some approaches described in the literature where the edge density is (approximately) fixed. We consider fixing the correlation threshold rather than the edge density to be more in accordance with our working hypothesis, where we expect a high and widespread correlation between the nodes during extreme seasons, which will be reflected in significant increases of the edge density and other metrics; it is the *change* of these metrics which will characterise extreme seasons.

In order to assess the impact of the time scales on the identification of extremes, we consider heat waves and droughts occuring in the summer half year (SHY, May to October) and summer season (June to August, JJA), so that the length of the time series for each year is 184 days and 92 days, respectively. Although droughts are known to occur also in winter, we only consider SHY and JJA droughts here. If we denote the number of nodes by $n$ and the edge density by $e$, the maximum possible number of edges is $e_{max} = \binom{n}{2} = n(n-1)/2$. The adjacency matrix $A$ is then an $n \times n$ matrix with $a_{ij} = 1$ if node $i$ and node $j$ are connected and 0 otherwise. The degree of node $k$, i.e. the number of nodes connected to it, will be denoted by $d_k$ and the average degree of the network will be denoted by $\bar{d}$. To analyse the adjacency matrix and to identify extremes, we considered the following metrics (see e.g. Newman (2003) or Donges et al. (2009)):

– the edge density $e$, defined as the number of edges in the network, divided by $e_{max}$; this can be considered a measure of the spatial extent and "connection strength" of the extreme event. To identify extremes, we will also use the normalised edge density, defined as $\epsilon = (e - \bar{e})/\sigma_e$ where $\bar{e}$ is the average over the years 1951 to 2019 and $\sigma_e$ is the corresponding standard deviation.

– the global (triangle) clustering coefficient $\bar{c}$, defined as the average of the local clustering coefficients $c_k = \Delta_k/\Delta_{max,k}$, where $\Delta_k$ is the number of triangles connected to node $k$ and $\Delta_{max,k} = \binom{d_k}{2}$ is the number of all triangles centered at node $k$ (see Newman (2003), Watts and Strogatz (1998)). Normalised values were calculated in the same way as for the edge densitiy.

– the distribution of the node degrees $d_k$, $k = 0 \cdots n-1$.

We found that in the framework of this study, these metrics and especially the edge densitiy are sufficient to identify extremes (see section 3) and therefore did not consider more elaborated metrics like path length, betweenness etc. as described e.g. in (Donges et al., 2009). As already mentioned, the time series of the yearly SHY and JJA metrics were normalised by their average and standard deviation over the period 1951 to 2019; if the normalised metric of a period is larger than one standard deviation, this period is considered extreme; values close to 1 (about $1 \pm 0.2$) are considered "border cases", possibly indicating moderate, small scale or short lived extremes.

## 2.2 Data used for building the CNs

Several time series of gridded temperature and precipitation data are freely available, e.g E-Obs (Cornes et al., 2018), ERA reanalyses (Hersbach et al., 2020) and data sets from national weather services, e.g. the German Weather Service DWD; differences between these data sets are due to spatial and temporal resolution, observations used and statistical/interpolation
methods. A comparison of such data sets can be found in (Skok et al., 2016).

In this study, we used the E-Obs V21.0e daily maximum temperature ($T_{max}$) and precipitation gridded daily data sets (https://surfobs.climate.copernicus.eu/dataaccess/access_eobs.php). This data set has a spatial resolution of 0.25 degrees and covers the period from 1950 to 2019; it is updated continuously. The selected region was 5 to 16 degrees longitude East and 47 to 56 degrees latitude North, covering Germany (henceforth called GE region, see Fig. 6). We selected E-Obs for its relatively
high resolution, its long time coverage and also for comparability due to its frequent use in other studies; but note that only data for land surfaces are provided by E-Obs. We focus here on Germany due to the high density of stations for interpolation and the availability of extreme event catalogues for comparison. For droughts, from the precipitation time series a 0-1 time series of dry days was calculated as follows: if for a given day, the daily precipitation sum was less than 1 mm, this day was a dry day and assigned a 1, otherwise, it was assigned a 0. Two nodes were connected if their correlation coefficient exceeded a
given threshold (see section 4). We adopted here the E-Obs definition of a dry day as a day with a daily precipitation sum less than 1 $mm/day$ (see https://www.ecad.eu/FAQ/index.php#5).

## 2.3 Identification of extreme events using EDI/EHI and other sources

There exist several indices to identify and to quantify the severity of extremes, like SPI, WASP index, SDI, PDI and several others for drought; they differ among each other in purpose, definition of extreme, method employed, spatial and temporal
scales, focus on meteorology (precipitation) or hydrology (soil moisture, runoff); a discussion of such differences for droughts can be found in Byun and Wilhite (1999). Therefore, each choice of index is somewhat arguable and mainly owned to the need for a reference.

In this study, extreme events are identified by using spatial (over the region considered) and temporal (over the season considered) averages of the effective drought index (EDI, Byun and Wilhite (1999)) and an analogous metric defined for heat,
the effective heat index (EHI, Sedlmeier et al. (2016)), which are basically time series of effective temperature and precipitation, normalised by mean and standard deviation. Therefore, (relative) extremes occur when these indices deviate markedly (usually one standard deviation) from zero. EDI and EHI are relatively easy to calculate, use a minimum of assumptions, need no correction for trends and take the memory effect of the soil and the atmosphere into account, which is important for the assessment of the severity of heat waves and droughts. Being aware that there is no "best index", we will also have a look at
other extreme event indices (see section 4).

We describe here the calculation of the effective drought index (EDI), the effective heat index (EHI) is calculated similarly by using $T_{max}$ (see Sedlmeier et al. (2016)). The EDI was proposed by (Byun and Wilhite, 1999) and describes drought extremes at a site as deviations from a climatological mean state; it uses the concept of effective precipitation (EP), which

takes the memory effect of the soil into account. It correlates highly with soil moisture, which makes it well suited for studying droughts.

The effective precipitation $EP$ for a given day $d$ is calculated as follows:

$$EP(d) = \sum_{k=1}^{365} \omega_k \cdot S_k(d)$$

where the weights are $\omega_k = 1/k, k = 1, \cdots, 365$ and

$$S_k(d) = \sum_{i=1}^{k} P(d-i)$$

is the precipitation sum over the last $k$ days before day $d$. From $EP(d)$, the (daily) $EDI(d)$ is calculated as

$$EDI(d) = (EP(d) - \overline{EP})/\sigma(EP)$$

where $\overline{EP}$ and $\sigma(EP)$ are the mean and standard deviation of $EP$ for SHY and JJA over the period 1951 to 2019.

An analogous measure can be defined for temperature, called the effective heat index (EHI) with the daily maximum temperature $T_{max}$ and $k = 49$ instead of $k = 365$ days. For the effective temperature, the value of 49 was determined as the lag where the autocorrelation function equals 0.5 (see Sedlmeier et al. (2016)).

One problem in connection with EDI/EHI and many other extreme indices is that they are defined at points, whereas for extremes, one is interested in area information. As mentioned in the introduction, this is one of the advantges of RCNs. For the comparison of the (area-wise) RCN metrics with the (point-wise) EDI/EHI, we calculated an area and seasonal average of the EDI from the area averaged effective precipitation and of the EHI from the area averaged effective $T_{max}$; to account for the smoothing of extremes due to this averaging, for a given year, season and region we define droughts as extreme when the spatially and temporally averaged EDI is less than -1, and heat events as extreme when the spatially and temporally averaged EHI is larger than +1. We are aware that there is a certain arbitrariness in this definition. We try to reduce this arbitrariness by considering also other indices when there are large differences between EDI/EHI and RCN metrics or by relaxing the threshold in cases where EDI/EHI or RCN metrics are close to the threshold ("border cases").

Valuable sources of information on the occurence of extremes are Hannaford et al. (2011), Parry et al. (2012) and Spinoni et al. (2015). Hannaford et al. (2011) provide a detailed analysis, based on precipitation and runoff observations, of drought events (meteorological and hydrological) for several regions in Europe, among them subregions of Germany for the period 1961 to 2005. We will refer mainly to this dataset to complement our comparison with EDI. For heat waves, we will refer to Kornhuber et al. (2019), Vautard et al. (2007),Vautard et al. (2020), Zschenderlein et al. (2019), Russo et al. (2015) and Luterbacher et al. (2004).

## 3   Sensitivity of the metrics to correlation thresholds

The choice of the correlation threshold of the time series determines the entries of the adjacency matrix, which characterises the network and determines all metrics like edge density, degree distribution, local and global clustering coefficient and other

derived metrics; it is the only adjustable parameter in our setup of the RCN. One can either fix the correlation threshold resulting in varying edge densities or fix the edge density by adjusting the correlation threshold, as done e.g in Weimer et al. (2016); both possibilities require a decision of which values to choose. We decided to use a fixed correlation threshold, since a high correlation above a fixed threshold on long (e.g. seasonal) time scales and over an extended area is an indication of a strong

persistent coupling between nodes, which is what we are looking for - we let the structure of the network reflect the given climatic situation. To see how the choice of the correlation threshold affects the metrics of the RCN, we conducted a series of sensitivity runs for drought and heat extremes. Essential criteria to judge the suitability are i) the edge density $e$ which should be not to small in order to have enough data for calculating the metrics, but also not too large in order to have a sufficiently large spread ($e$ values in the literature are in the order of 0.1), and ii) the ability of the network to detect significant differences

between normal and extreme years. We varied the correlation threshold $\rho_0$ for $\rho_0 = 0.70, 0.80, 0.85, 0.90, 0.95, 0.99$ for the GE region over the years 1951 to 2019. The results are discussed now separately for drought and heat.

## 3.1   Sensitivity droughts

The left hand part of Fig. 1 shows the variation of edge density $e$ (blue circles) and global clustering coefficient $\bar{c}$ (red circles), all averaged over the summer half year for the years 1951 to 2019, with the correlation thresholds defined above. Also shown

is the spread, i.e. ratio standard deviation/average for $e$ (blue triangles) and $\bar{c}$ (red triangles) as a function of the correlation threshold. As expected, $e$ and $\bar{c}$ decrease considerably with increasing $\rho_0$; however, the sensitivity of $\bar{c}$ is much less pronounced, although $e$ and $\bar{c}$ are highly correlated for all $\rho_0$. For $\rho_0 = 0.70$, the edge density is very high, but the spread is low. The other extreme occcurs for $\rho_0 = 0.99$: the spread is sufficiently large, but the values are based on too few connections, so the statistics are not reliable (for $\rho_0 = 0.99$, there are only about 90 edges out of almost 900,000 on average). For $\rho_0 = 0.85, 0.90, 0.95$, edge

densities are around 0.1, which is in the range used for so-called sparse networks in the literature (Radebach et al., 2013), and the ratio spread/average is around 0.5, which we consider sufficiently large.

The right hand part of Fig. 1 shows the ratio $q$ of the edge density (blue dots) and global clustering coefficient (red dots) averaged over extreme years (defined as years with $\epsilon > 1$) to those averaged over normal years (defined as years with $|\epsilon| < 0.3$). High values of this ratio indicate that there is a significant difference between extreme and normal years - the ability of the

RCN we are looking for. The ratio $q$ is low for $\rho_0 = 0.70$ due to the small spread. High values above 1.6 are attained for $\rho_0$ in the range 0.8 to 0.95. A Wilcoxon test indicates that these differences between normal and extreme seasons are significant above the 99 % level. Concerning the global clustering coefficient $\bar{c}$, the ratio between extreme to normal years (red dots) is only slightly above 1, i.e it does not discriminate well between normal and extreme years which makes it less suitable for extreme detection.

From these results, we infer that suitable values of $\rho_0$ for extreme drought detection are between 0.85 and 0.95, and a good separation between normal and extreme years can be achieved using the (normalised) edge density as a metric. We will use this metric and the range of $\rho_0 = 0.85, 0.90, 0.95$ for the comparison of the RCN method with data from the literature in section 4.

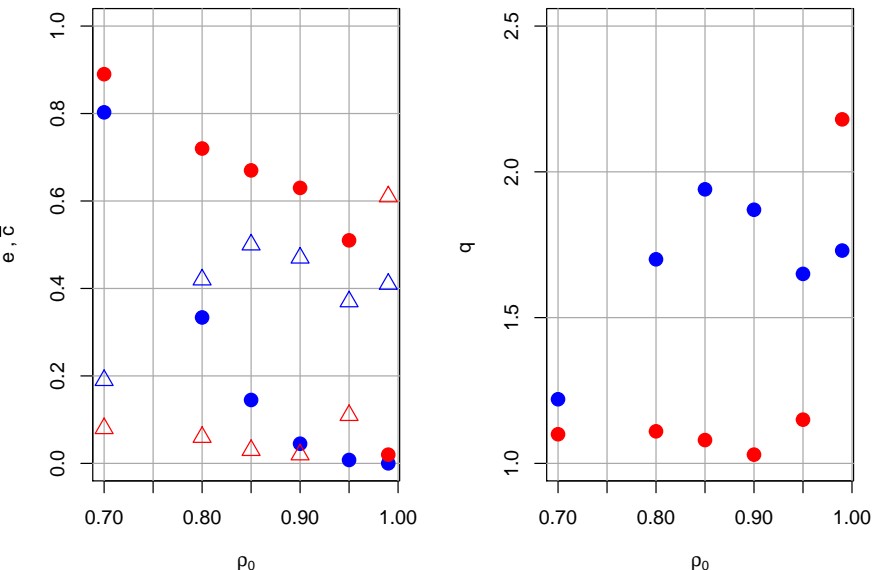

**Figure 1.** Drought GE SHY, left: average (period 1951-2019) edge density $e$ (blue dots) and global clustering coefficient $\bar{c}$ (red dots) as a function of the correlation threshold $\rho_0$ for SHY Germany; also shown is the ratio of standard deviation to average for $e$ (blue open triangles) and $\bar{c}$ (red open triangles) as a function of the correlation threshold. Right: the same for the ratio $q$ of extreme to normal years, calculated with $e$ (blue) and $\bar{c}$ (red).

| $\rho_0$ | 0.85 | | 0.90 | | 0.95 | |
|---|---|---|---|---|---|---|
| normal years | $\bar{d}$ | $d_{max}$ | $\bar{d}$ | $d_{max}$ | $\bar{d}$ | $d_{max}$ |
| 1953 | 190 | 446 | 60 | 172 | 10 | 40 |
| 1970 | 175 | 366 | 57 | 146 | 10 | 28 |
| 1980 | 168 | 310 | 57 | 134 | 11 | 43 |
| 1994 | 190 | 391 | 58 | 116 | 11 | 37 |
| extreme years | $\bar{d}$ | $d_{max}$ | $\bar{d}$ | $d_{max}$ | $\bar{d}$ | $d_{max}$ |
| 1959 | 619 | 1085 | 187 | 509 | 25 | 70 |
| 1976 | 354 | 750 | 106 | 218 | 18 | 41 |
| 2003 | 376 | 760 | 105 | 222 | 16 | 50 |
| 2018 | 525 | 1034 | 159 | 428 | 21 | 101 |

**Table 1.** Drought GE SHY: average and maximum degrees for $\rho_0 = 0.85, 0.90, 0.95$ for four normal years (1953, 1970, 1980 and 1994) and four extreme years (1959, 1976, 2003 and 2018).

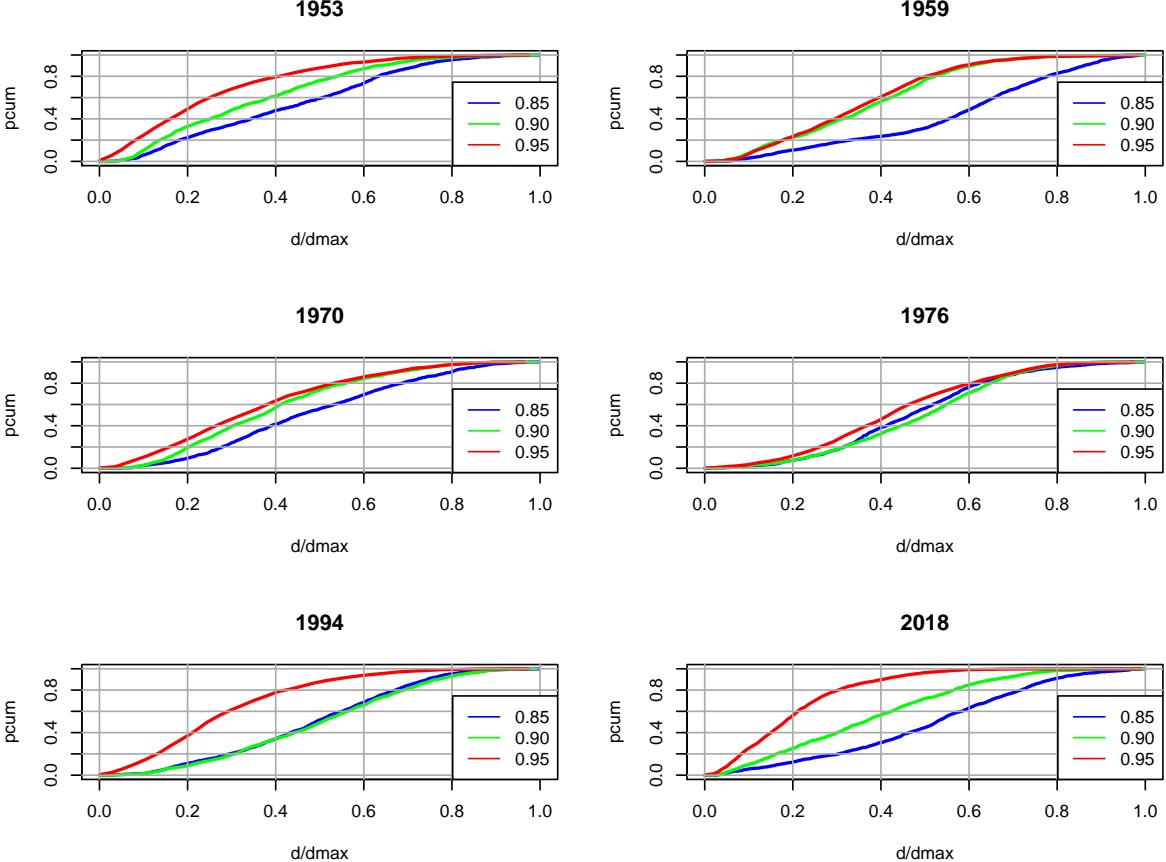

**Figure 2.** Drought GE SHY: dependence of the cumulative distribution of the node degrees for three normal (1953, 1970 and 1994, left) and three extreme (1959, 1976 and 2018, right) years on the correlation threshold $\rho_0$.

Table 1 shows the average and maximum degrees for $\rho_0 = 0.85, 0.90, 0.95$ for four normal years (1953, 1970, 1980 and 1994) and four extreme years (1959, 1976, 2003 and 2018). The average degrees decrease like the edge density (see Fig. 1) with increasing $\rho_0$ for normal as well as extreme years, and do not vary much between the normal years; for extreme years, the spread is larger. For all $\rho_0$, the average as well as the maximum degree increase considerably from normal to extreme years by a factor of about two to three; approximately the same factor applies to the ratio maximum to average degree. Thus, the overall behavior of the degree distributions is the same for the $\rho_0$ values presented and is similar within the normal and extreme year groups. The cumulative distribution of the node degrees for GE SHY is shown in Fig. 2 exemplarily for three normal (1953, 1970 and 1994) and three extreme (1959, 1976 and 2018) years, again for $\rho_0 = 0.80, 0.90, 0.95$; for each year, the degrees are normalised with the maximum degree for better comparison. Roughly, two kinds of distribution can be discerned: i) more asymmetric distributions with either a pronounced maximum at low or high degrees (years 1953 and 2018 for $\rho_0 = 0.95$) and

ii) more symmetric (years 1976 and 1994 for $\rho_0 = 0.85, 0.90$), flat distributions with many low degree as well as high degree nodes and a less pronounced maximum. However, there are many years which cannot attributed clearly to either type and both types can appear in normal as well as in extreme years. As Figure 1 and Table 1 showed, the main difference between normal and extreme years is the higher average (i.e. edge density) and maximum degree which are considerably higher during extreme

years.

We also found that whereas during extreme years, the distribution of the node degrees is more uniform and has more high-degree nodes, the distribution during normal years often resembles a Poisson distribution with parameter $\lambda = \overline{d_k}$ (the average node degree), which is characteristic for random networks (Newman, 2003). This could be an indication of the presence of

a random or, in our case, a random geometric graph (Penrose, 2003), (Ferrero and Gandino, 2017). A comparison between the probability distributions for the normal year 2013 and the extreme year 2018 is shown in Fig. 3 as an example. A more Poisson-like distribution during normal years could be explained by the higher level of "noise" induced by the higher variability of weather systems during normal years and the presence of complex orography and varying land use which disturb the organisation process and thus lead to lower correlations and lower edge densities. A detailed study to substantiate this observation is

beyond the scope of this paper.

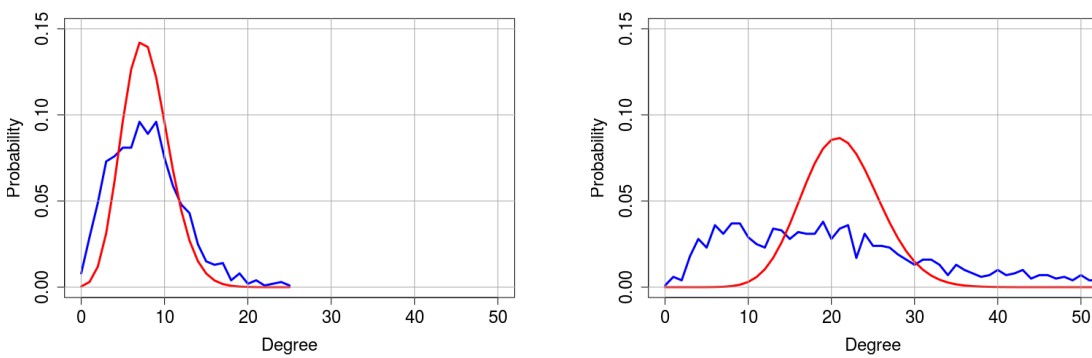

**Figure 3.** Node degree probability distribution for SHY droughts in the GE region for the normal year 2013 (left) and the extreme year 2018 (right). Blue: distribution according to the RCN, red: Poisson distribution with parameter $\lambda = \overline{d_k}$ (the average node degree).

### 3.2 Sensitivity heat waves

Figure 4 is the same as Fig. 1, but for heat waves instead of droughts: the left hand part shows the variation of edge density $e$ and global clustering coefficient $\bar{c}$, again averaged over the years 1951 to 2019, with the correlation threshold. We chose here the summer months (JJA) since these months turned out to be more suitable to identify heat waves (see section 4.2).

Again, edge density and clustering coefficient decrease considerably with increasing $\rho_0$. Except for $\rho_0 = 0.70$, edge densities

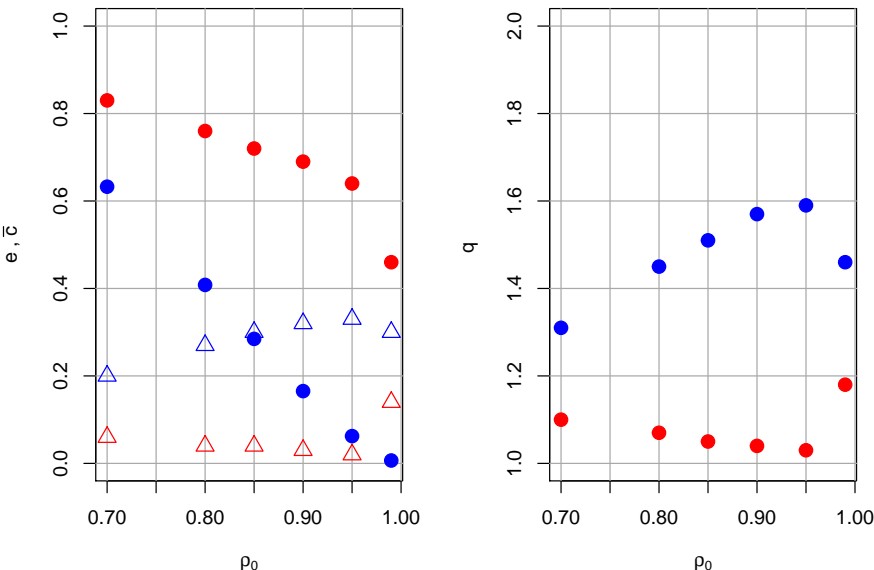

**Figure 4.** Same as Fig. 1,but for heat waves GE JJA.

are higher than for the drought case; this could be due to the fact that correlations are higher for the continuously varying daily maximum temperatures compared to the 0-1 time series for droughts. Again, for $\rho_0 = 0.70$ edge density and clustering coefficient are high, but the spread is low, and for $\rho_0 = 0.99$ the spread is sufficiently large, but there are few connections, making the statistics unreliable. As for droughts, we calculated the ratio $q$ of the average edge density for extreme years (defined as

normalised edge density $\epsilon > 1$) to the edge density averaged over normal years (defined as $|\epsilon| < 0.3$ ). High values around 1.5 to 1.6 are attained in the range 0.85 to 0.95. According to the Wilcoxon test, the differences between normal and extreme seasons are significant above the 99 % level. As for droughts, the ratio extreme to normal years is only slightly above 1 for the global clustering coefficient, i.e it does not discriminate well between normal and extreme years which makes it less suitable for extreme heat wave detection.

Table 2 shows the average and maximum degrees for $\rho_0 = 0.85, 0.90, 0.95$ for four normal years (1975, 1991, 2005 and 2009) and four extreme years (1978, 2003, 2006 and 2013). The average degrees decrease like the edge density (see Fig. 4) with increasing $\rho_0$ for normal as well as extreme years, and do not vary much between the years. All values are higher than in the drought case, and the differences normal to extreme are smaller. For all $\rho_0$, the average as well as the maximum degree increase from normal to extreme years by about 50 to 100 %; approximately the same factor applies to the ratio maximum to average

degree. Thus, the overall behavior of the degree distributions is the same for the $\rho_0$ values presented and is similar within the normal and extreme year groups. The cumulative distribution of the node degrees for GE JJA is shown in Fig. 5 exemplarily

for three normal (1975, 1991 and 2005) and three extreme (1978, 2003 and 2006) years, again for $\rho_0 = 0.80, 0.90, 0.95$. As for droughts, two kinds of distribution can be discerned: i) more asymmetric distributions with either a pronounced maximum at low or high degrees, and ii) more symmetric, flat distributions with many low degree as well as high degree nodes and a less pronounced maximum. Compared to the drought case, the latter kind of distribution is the more frequent one, but still, there are

5 many years which cannot attributed clearly to either type and both types can appear in normal as well as in extreme years. Like for droughts, the main difference between normal and extreme years is the higher average (i.e. edge density) and maximum degree which are considerably higher during extreme years.

| $\rho_0$ | 0.85 | | 0.90 | | 0.95 | |
|---|---|---|---|---|---|---|
| normal years | $\bar{d}$ | $d_{max}$ | $\bar{d}$ | $d_{max}$ | $\bar{d}$ | $d_{max}$ |
| 1975 | 370 | 676 | 221 | 373 | 84 | 153 |
| 1991 | 389 | 649 | 227 | 354 | 87 | 188 |
| 2005 | 379 | 676 | 215 | 407 | 80 | 153 |
| 2009 | 389 | 720 | 225 | 421 | 84 | 180 |
| extreme years | $\bar{d}$ | $d_{max}$ | $\bar{d}$ | $d_{max}$ | $\bar{d}$ | $d_{max}$ |
| 1978 | 581 | 1010 | 348 | 681 | 132 | 261 |
| 2003 | 534 | 970 | 340 | 533 | 141 | 312 |
| 2006 | 643 | 1036 | 395 | 677 | 157 | 308 |
| 2013 | 507 | 919 | 310 | 537 | 122 | 246 |

**Table 2.** Same as Table 1, but for heat waves GE JJA.

From these findings, we conclude that for the detection of heat waves, suitable values of $\rho_0$ are between 0.85 and 0.95. This

is the same range as for droughts, so at least for heat and drought, no adjusting of the threshold $\rho_0$ is necessary. We will use these values in the next section, where we compare the RCN results with data from the literature.

We can summarise the findings of this sensitivity study as follows: thresholds $\rho_0$ between 0.85 and 0.95 give reliable results in terms of extreme detection and significance of the statistics for both drought and heat waves. At least for these extremes, no

adjusting of the threshold is necessary. The exact value of $\rho_0$ seems less important. Node degrees and edge densities are higher for heat waves than for droughts. The global clustering coefficient $\bar{c}$, although highly correlated to $e$ for all $\rho_0$, discriminates not well between extreme and normal years. In the remaining sections of this paper, we will therefore use the (normalised) edge density as metric to detect extremes and vary $\rho_0$ for values $\rho_0 = 0.85, 0.90, 0.95$.

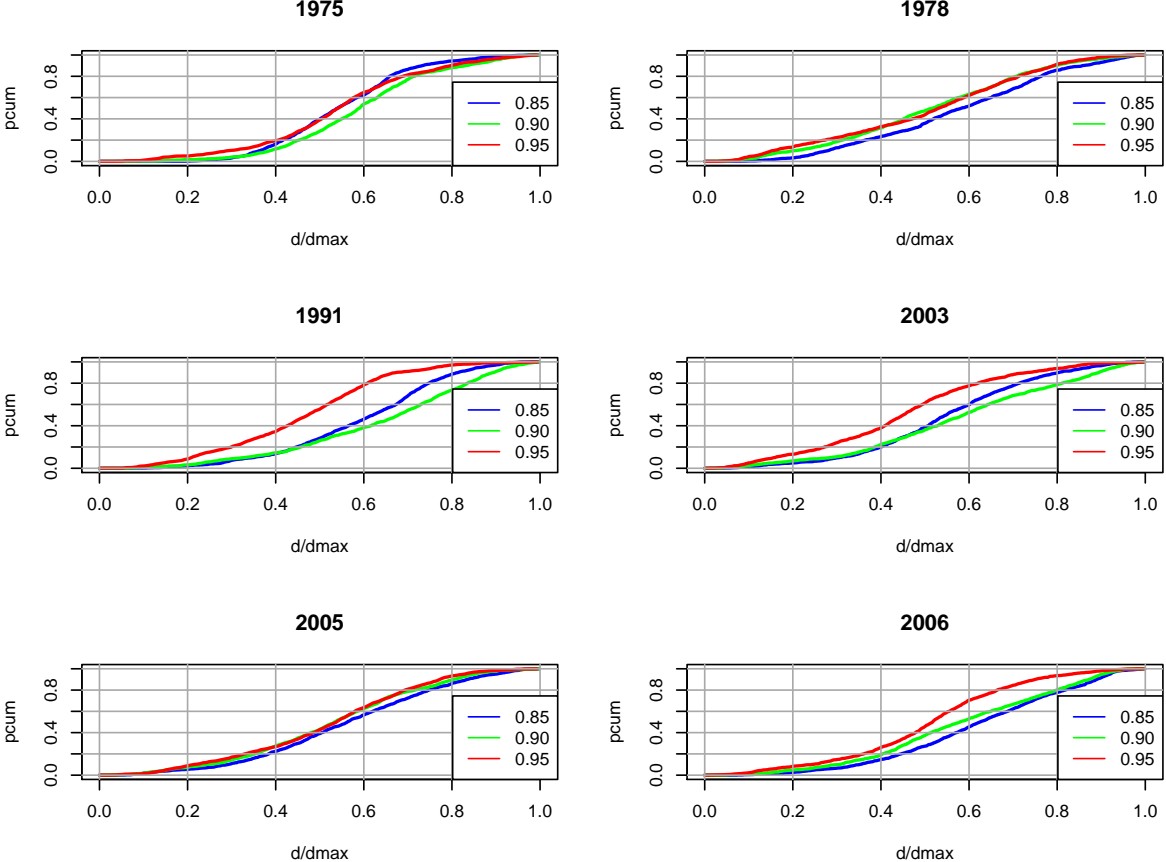

**Figure 5.** Same as Fig. 2, but for heat waves GE JJA. Normal years are 1975, 1991 and 2005 (left), extreme years are 1978, 2003 and 2006 (right).

## 4  Comparison of the RCN results with other extreme indices (mainly EDI and EHI)

In this section, we discuss the comparison between EDI/EHI and RCN edge density for the summer half years (SHY, May till October) and summer seasons (JJA, June till August) for Germany (GE) and the two subregions northern Germany (GEN) and southern Germany (GES) with respect to droughts and heat waves during the period 1951 - 2019. EDI and EHI are averaged 5 spatially over the respective regions and temporally over the respective season. For the reasons given in the previous section, we only consider the normalised edge density $\epsilon$ as RCN metrics. Extremes are defined as $\epsilon > 1$ for the RCN and as $EDI < -1$ and $EHI > 1$. (Remark: values $\epsilon < -1$ would mean that the edge density is considerably below average; this could be caused either by wet/cool years or by a low correlation due to uncorrelated small scale events. Both possibilities are not a focus of this study). For comparison, we give the results for the correlation thresholds $\rho_0 = 0.85, 0.90, 0.95$ as discussed in the previous 10 section. The GE, GEN and GES regions are shown in Fig. 6.

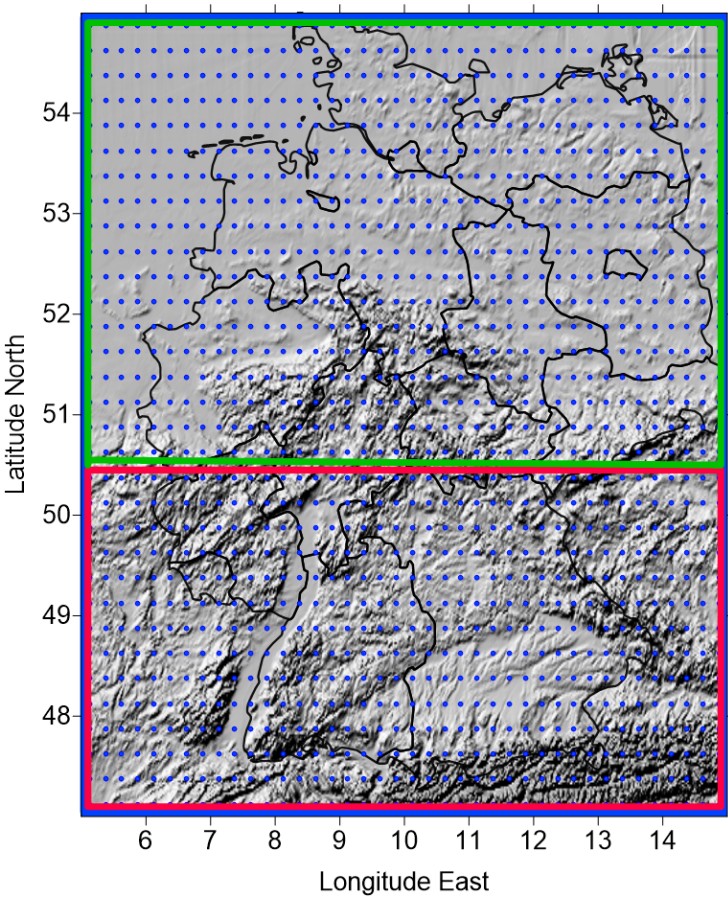

**Figure 6.** Relief map of Germany with the GE (blue frame) and the subregions southern Germany GES (red frame) and northern Germany GEN (green frame) considered in this study. The E-Obs grid is marked by blue dots.

### 4.1 Droughts: GE SHY

Table 3 shows the years which are identified as extreme by the RCN or by the EDI (according to the definitions above) for the summer half years over Germany for the values of $\rho_0$ indicated above. Six years are identified by both EDI and RCN as years with extreme droughts, namely 1959, 1964, 1976, 1991, 2003 and 2018. These years are also identified as extreme in the literature (e.g. Spinoni et al. (2015), Hannaford et al. (2011)), and also the European Drought Reference (EDR) Database (https://www.geo.uio.no/edc/droughtdb/edr/DroughtEvents.php), so all extreme years are found by the RCN.

The year 1973, identified extreme by EDI, is just below the RCN threshold of $\epsilon > 1$. The years 1969, 1986,1989,1990 are not deemed extreme in EDI, whereas these years are identified as moderately extreme in parts of Germany in Hannaford et al. (2011); the combination of weaker signal and only regional occurence could be a reason for the non-detection by EDI.

Thus, we can state that the RCN is able to detect the severe and moderately severe SHY drought events quoted in the literature

including less severe or only regionally severe years.

| year | 0.85 | 0.90 | 0.95 | EDI |
|------|------|------|------|-----|
| 1959 | o | o | o | o |
| 1964 | o | o | o | o |
| 1969 | o | - | o | - |
| 1973 | - | - | - | o |
| 1976 | o | o | o | o |
| 1986 | o | o | o | - |
| 1989 | o | o | o | - |
| 1990 | - | o | o | - |
| 1991 | o | o | o | o |
| 2003 | o | o | o | o |
| 2018 | o | o | o | o |

**Table 3.** Comparison of extreme drought summer half years (SHY) between 1951 and 2019 as identified by EDI and the RCN. Bars indicate SHYs identified as not extreme, open circles indicate SHYs identified as extreme. Years according to EDI are shown in the rightmost column, years according to the RCN based on the normalised edge density for the correlation thresholds $\rho_0 = 0.85, 0.90, 0.95$ are shown in columns two to four.

### 4.1.1 Droughts: RCN metrics differences between normal and extreme years

To illustrate the differences in the network metrics between normal and extreme years, we calculate the ratio $q$ of the edge density averaged over extreme years between 1959 and 2019 (defined as $\epsilon > 1$) to the edge density averaged over normal years (defined as $|\epsilon| < 0.3$). This ratio, together with the edge density, is shown for GE in the second and third column of Table 4. The edge density decrases from 0.15 to 0.01. The value of $q$ is almost 2 for $\rho_0 = 0.85, 0.90$; this difference between extreme and normal years is significant at the 99% level according to a Wilcoxon test and shows that the RCN is clearly able to differentiate between normal and extreme years.

Table 4 also makes differences between Germany as a whole and the two subregions evident. All $q$ values are quite high, with the more flat and homogeneous GEN having generally higher values than GES which is affected more by orographical "noise" (see next section).

The spatial distributions of the network metrics also differ considerably between normal and extreme years. An example is shown in Fig. 7, which shows the spatial node degree distribution for the normal year 1970 (left) and the extreme year 1976 (right) for $\rho_0 = 0.95$. In 1976, average degree, maximum degree and edge density are almost double the values of 1970. Also regional differences (cf. Table 4) become visible: in the flat northern parts of Germany, especially in the northeast with

| $\rho_0$ | GE | | GEN | | GES | |
|---|---|---|---|---|---|---|
| | $\bar{e}$ | $q$ | $\bar{e}$ | $q$ | $\bar{e}$ | $q$ |
| 0.85 | 0.15 | 1.98 | 0.24 | 2.36 | 0.27 | 1.54 |
| 0.90 | 0.05 | 1.90 | 0.08 | 2.49 | 0.09 | 1.58 |
| 0.95 | 0.01 | 1.62 | 0.01 | 2.12 | 0.02 | 1.46 |

**Table 4.** Dependence of the average edge density $\bar{e}$ and the ratio $q = \bar{e}_{extr}/\bar{e}_{norm}$ (edge density during extreme years to edge density during normal years) over the period 1951-2019 on the correlation threshold ($\rho_0 = 0.85, 0.90, 0.95$). Results are shown for droughts during SHY in the GE, GEN and GES regions.

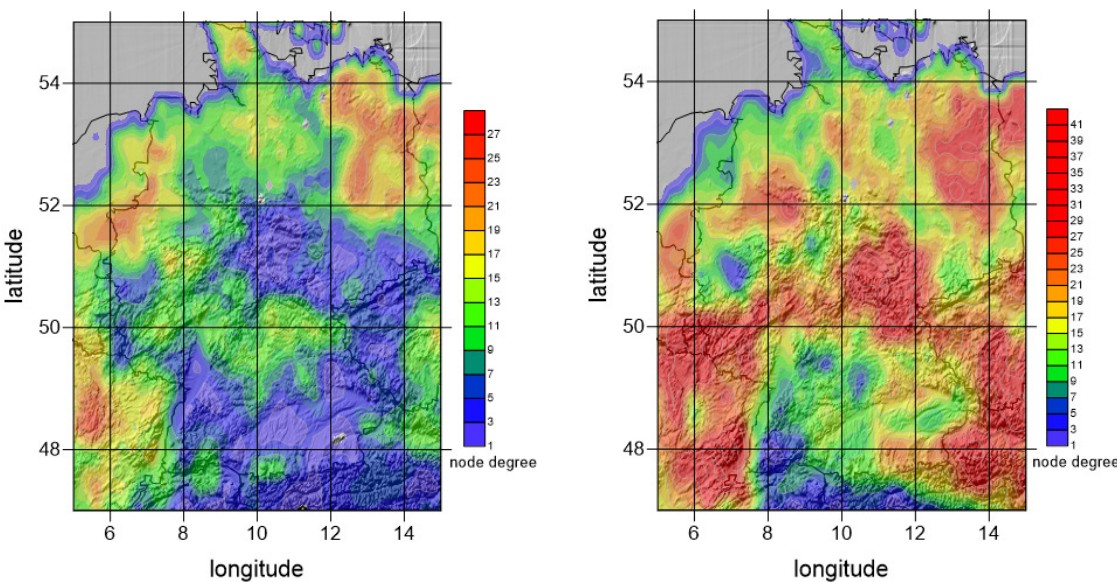

**Figure 7.** Comparison of the spatial distribution of the node degree between the normal year 1970 (left) and the extreme year 1976 (right). Note the different scales.

quite uniform sandy soils (reducing the precipitation recycling rate), node degrees tend to be considerably higher than in the more rugged, moutaineous and forested southern parts which favor irregular precipitation distribution and thus act as noise in the adjacency matrix calculation. Mountaineous regions are also often the ones with lower node degree in the 1976 extreme drought year; however, exceptions occur in some mountaineous regions in the North and East, perhaps due to stronger impact of blocking highs, increased continentality in the East and less available moisture in the atmosphere.

#### 4.1.2 Droughts: SHY extremes in GEN and GES

To illustrate the effect of orography and geographical situation, we compare the identified extreme years of the GEN region with the ones of the mountaineous GES region of Germany (see Fig. 6). Whereas GES has a complex mountaineous orography with varying land use, GEN is mostly flat, has more uniform land use with dominant sandy soils and has in its eastern parts a more continental climate. It is known that there are differences in the occurence and intensity of extreme droughts within Germany (see e.g. Samaniego et al. (2013)). Droughts can be quite regional and can occur in different years in the northern and northeastern parts of Germany than in the southern parts. To see if this is reflected in EDI and the RCN data, we compare the EDI and the RCN edge density $\epsilon$ for the GES and GEN subregions.

| year | 0.85 | 0.90 | 0.95 | EDI |
|------|------|------|------|-----|
| 1959 | o | o | o | o |
| 1973 | - | - | - | o |
| 1976 | o | o | o | o |
| 1986 | - | - | o | - |
| 1989 | o | o | o | o |
| 1990 | - | - | o | - |
| 1992 | o | o | o | - |
| 1996 | - | - | - | o |
| 2003 | o | o | - | - |
| 2018 | o | o | o | o |

**Table 5.** As Table 3, but for drought GEN SHY.

Table 5 shows the results of the RCN for the different $\rho_0$ and EDI for GEN. EDI identifies the six years 1959, 1973, 1976, 1989, 1996 and 2018 as extreme. EDI and RCN agree in the five years 1959, 1976, 1989 and 2018. In the years 1973 and 1996, extreme years in EDI, EDI is just below the threshold (value -1.03), so these years could be considered as "border case" years. On the other hand, for the years 1992 and 2003, identified as extreme by the RCN, the edge density is just above the threshold, so these years represent RCN "border cases". In view of this, we can say that there is very good agreement between EDI and RCN (and also with the literature) for GEN. The years 1964 and 1991 do not appear as extremes for GEN, but do so for GE; this indicates that regional differences can be accounted for.

Table 6 shows the results for GES. EDI identifies the seven years 1964, 1971, 1976, 1991, 2003, 2015 and 2018 as extreme. The six years 1964, 1971, 1976, 1991, 2003, and 2018 are identified as extreme years in GES by both RCN and EDI. The year 2015, an additional extreme year in EDI just below the threshold, is not found by the RCN. For GES, RCN but not EDI identifies the five years 1959, 1962, 1969, 1986 and 1997 as extreme drought years. Of these, the years 1962, 1969 and 1997 are also drought years in Hannaford et al. (2011). There are interesting differences in the occurence of extreme years between GES and GEN. For example, the year 1989, an extreme year in GEN, does not appear in GES, whereas the year 1991 is extreme in

| year | 0.85 | 0.90 | 0.95 | EDI |
|------|------|------|------|-----|
| 1959 | o | o | o | - |
| 1961 | - | - | o | - |
| 1962 | o | o | o | - |
| 1964 | o | o | o | o |
| 1969 | o | o | o | - |
| 1971 | o | o | o | o |
| 1976 | o | o | o | o |
| 1986 | o | o | o | - |
| 1989 | - | o | - | - |
| 1990 | - | - | o | - |
| 1991 | o | o | o | o |
| 1997 | o | o | o | - |
| 2003 | o | o | o | o |
| 2005 | o | o | - | - |
| 2015 | - | - | - | o |
| 2018 | o | o | o | o |
| 2019 | o | - | - | - |

**Table 6.** As Table 3, but for drought GES SHY.

GES, but not in GEN. These regional differences, which can be seen in the maps in Samaniego et al. (2013), are well captured by the RCN and indicate that the RCN is able to identify droughts at varying spatial scales. They also illustrate the fact that the spatial scales of droughts can be down to the order of one hundred kilometers.

### 4.1.3 Droughts: GE JJA

5  For hydrology and agriculture it is of interest to know on shorter time scales when droughts are to be expected. It is also interesting to see how the RCN behaves on shorter time scales. We therefore compared the appearance of droughts obtained with EDI with ones obtained by the RCN for the summer (JJA) months.

For JJA, the ratio $q$ of extreme to normal years is above 2 and thus considerably higher than for SHY (not shown). This may be due to the fact that extreme droughts occur predominantly during the JJA months and are therefore better captured in this

10  shorter time window. Table 7 compares droughts derived from the RCN for JJA with the corresponding results obtained with EDI. The Table shows again good agreement between EDI an RCN. EDI identifies the seven years 1959, 1964, 1973, 1976, 2003, 2015 and 2018 as extreme. Both EDI and RCN identify the five drought years 1964, 1973, 1976, 2003 and 2018, so all years identified by EDI, except 1959 and 2015, are also identified by the RCN; the latter two years are border case with EDI just below -1.

| year | 0.85 | 0.90 | 0.95 | EDI |
|------|------|------|------|-----|
| 1959 | o | - | - | o |
| 1964 | o | o | o | o |
| 1973 | o | o | o | o |
| 1976 | o | o | o | o |
| 1983 | o | o | o | - |
| 2003 | o | o | o | o |
| 2013 | o | o | o | - |
| 2015 | - | - | - | o |
| 2018 | o | o | o | o |
| 2019 | o | - | - | - |

**Table 7.** As Table 3, but for drought GE JJA.

## 4.2 Heat waves

In this section, we apply our RCN to heat waves and compare the RCN metrics with the EHI for Germany for the summer half year (SHY) and the summer season (JJA), respectively. Like for droughts, we present the results for $\rho_0 = 0.85, 0.90, 0.95$ and use the edge density as the relevant metric.

5  ### 4.2.1 Heat waves: GE SHY

| year | 0.85 | 0.90 | 0.95 | EHI |
|------|------|------|------|-----|
| 1952 | o | o | o | - |
| 1964 | o | o | o | - |
| 1974 | o | o | o | - |
| 1983 | - | - | o | - |
| 1991 | o | o | o | - |
| 1992 | o | o | o | - |
| 1994 | o | o | o | - |
| 2003 | o | o | o | o |
| 2006 | - | - | - | o |
| 2009 | - | o | - | - |
| 2010 | o | o | o | - |
| 2015 | o | o | o | - |

**Table 8.** As Table 3, but for heat waves GE SHY.

Table 8 shows the SHY years identified as extreme either by the RCN ($\epsilon > 1$) or by EHI ($EHI > 1$) for GE and the years between 1951 and 2019. The years 2003 and 2006 are classified as extreme by the EHI, in line with literature (Luterbacher et al. (2004), Russo et al. (2015)), but only the year 2003 is an extreme year for RCN. The literature lists several years with extreme heat events in Germany, namely 1976, 1983, 1994, 1995, 2010, 2013 and 2015 (Vautard et al. (2007), Vautard et al. (2020), Kornhuber et al. (2019), Zschenderlein et al. (2019)); these years are not identified as extreme by the EHI for SHY, but three of them (1994, 2010 and 2015) are identified by the RCN consistently for all $\rho_0$ values. On the other hand, some years are identified as extreme by the RCN (1952, 1964, 1974, 1991 and 1992) which are not recorded as extreme in the literature, and the extreme year 2006 is not detected.

### 4.2.2  Heat waves: GE JJA

| year | 0.85 | 0.90 | 0.95 | EHI |
|------|------|------|------|-----|
| 1952 | -    | -    | o    | -   |
| 1957 | o    | -    | -    | -   |
| 1969 | o    | o    | o    | -   |
| 1976 | -    | -    | o    | -   |
| 1978 | o    | o    | o    | -   |
| 1980 | o    | o    | -    | -   |
| 1983 | o    | o    | o    | -   |
| 1986 | o    | o    | o    | -   |
| 1994 | o    | o    | o    | -   |
| 1995 | o    | o    | o    | -   |
| 2003 | o    | o    | o    | o   |
| 2006 | o    | o    | o    | o   |
| 2013 | o    | o    | o    | -   |
| 2015 | -    | -    | -    | o   |
| 2018 | -    | -    | -    | o   |

**Table 9.** As Table 3, but for heat waves GE JJA.

The results of section 4.2.1 show that there are discrepancies between heat events listed in the literature and heat events identified by the RCN. One reason could be that the averaging period (SHY) is too long to identify heat events in GE. For this reason, we look at a shorter averaging period in this section, namely JJA. As in SHY, the metrics in JJA are highly consistent. In contrast to SHY, in JJA EHI identifies four severe heat events (compared to two in SHY), namely in the years 2003, 2006, 2015 and 2018 (Table 9), in accordance with the literature (Kornhuber et al., 2019). Of these, RCN detects the years 2003 and 2006. Whereas the heat event of 2006 is now detected, the 2015 event, detected for SHY, is now missed. The years 1983, 1994, 1995 and 2013 are identified as heat events by all thresholds of the RCN, in line with literature (Vautard et al. (2007), Russo

et al. (2015), Zschenderlein et al. (2019), Vautard et al. (2020)). However, the years 1969, 1978 and 1986 are listed as heat events by the RCN; for these, there is no indication as regionally and seasonally extended extreme events in the literature. In view of these results we can state that shorter periods improve the detection rate (six out of eight events are detected, three events are falsely detected), but the detection rate for droughts is considerably better than the one for heat waves. The reasons for this are not clear; since the network results are quite consistent in themselves (dependence on $\rho_0$, high correlation among network metrics), improvements could be achieved by changing the construction of the adjacency matrix, e.g. by using different similarity measures as outlined in section 3. However, such a study is beyond the scope of the present paper.

### 4.2.3 Heat waves: RCN metric differences between normal and extreme heat years in GE, GES and GEN

In order to investigate the impact of orography and geographical situation on extreme heat events, the edge density $\bar{e}$ and the ratio $q = \bar{e}_{extr}/\bar{e}_{norm}$ (extreme/normal years) are compared for GE and the GES and GEN subregions. Table 10 shows this comparison. The $q$ values show that the edge densities increase considerably in extreme years compared to normal years, which indicates a clear separation between normal and extreme years. In contrast to droughts, there is no marked metrics difference between GEN and GES. It is interesting to observe that the edge densities are larger for the subregions than for GE, which could be an indication of GEN and GES belonging to different communities (e.g. Newman (2019)); this is however speculative and would require a detailed study. As already discussed in section 3, heat wave edge densities are considerably higher than drought edge densities, especially in the GEN and GES subregions.

|  | GE |  | GEN |  | GES |  |
| --- | --- | --- | --- | --- | --- | --- |
| $\rho_0$ | $\bar{e}$ | $q$ | $\bar{e}$ | $q$ | $\bar{e}$ | $q$ |
| 0.85 | 0.28 | 1.51 | 0.42 | 1.51 | 0.47 | 1.38 |
| 0.90 | 0.16 | 1.57 | 0.26 | 1.66 | 0.30 | 1.50 |
| 0.95 | 0.06 | 1.59 | 0.10 | 1.59 | 0.12 | 1.61 |

**Table 10.** Heat waves JJA: comparison of edge density and ratio $q = \bar{e}_{extr}/\bar{e}_{norm}$ (extreme/normal years) for GE, GEN,GES and correlation threshold $\rho_0 = 0.85, 0.90, 0.95$.

## 5 Summary

We used Regional Climate Networks (RCNs) to identify heat waves and droughts in Germany and two subregions for the summer half years (SHY, May-October) and summer seasons (JJA, June-August) during the period 1951 to 2019. The RCNs were constructed from maximum daily temperature and precipitation data, respectively, on the regular 0.25 degree grid of the EObs data set. The season-wise correlation of time series of these daily data was used to construct the adjacency matrix of the network. Nodes were connected by an edge if the Pearson correlation coefficient of the time series was above a fixed threshold

$\rho_0$. Candidate metrics to identify extremes were the edge density $\epsilon$ and the average clustering coefficient $\bar{c}$, which turned out to be highly correlated. A sensitivity study showed that $\rho_0 = 0.85, 0.90, 0.95$ together with the edge density as metric gave reasonable results. The extreme indices for comparison were the effective drought (EDI) and heat (EHI) index respectively, based on the same time series, and complemented by other published event catalogues.

Our results show that the RCNs are able to identify extremes and also to distinguish, to a certain extent, between severe and moderate events. For droughts, there is a very good agreement between EDI and RCN results. The results for heat waves, although giving reasonable agreement, are less satisfactory than the ones for droughts: some events are not detected, while others are detected, but not identified as extreme neither by EHI nor elsewhere in the literature. Reasons could be that some events are too local, too short lived, are centered outside the regions considered, are only in the season considered or are

not intense enough. It could also be necessary to construct the adjacency matrix of the network differently either by using a different statistical association measure, e.g. event synchronisation. Finding the reasons for the disagreement would require a detailed analysis of the regional and temporal temperature and precipitation conditions in the respective years, which is beyond the scope of the present paper.

Varying the size of the region considered showed that the occurence of extreme events found by the RCN varies with the region,

in accordance with observations. Furthermore, it turned out that the applicability of RCNs to identify summertime heat events depends on the averaging period; this dependence is much less for droughts, probably due to the longer time scales. All metrics increase significantly during extreme events. Degree probability distributions vary considerably between more flat uniform ones and those with pronounced maxima, but cannot be attributed to normal or extreme years. An interesting observation is that for normal years, the distribution of the node degrees often resembles a Poisson distribution, characteristic of random

networks, while for extreme years the distribution is more uniform.

There are several advantages of RCNs over conventional methods: they provide information for whole areas (in contrast to the point-wise information from standard indices) and the extent of affected areas, they can be applied to arbitrary regions, the underlying nodes can be distributed arbitrarily, they are easy to construct and they provide details otherwise difficult to avail of (e.g. regional and seasonal differences, vulnerable regions and impact of orography). An additional advantage of the method is

that it is very fast, which makes it suitable for postprocessing climate model data. The RCN for Germany had 1338 nodes, i.e. an adjacency matrix with about 1.8 million entries; a run takes less than 4 seconds per year on a laptop, i.e. less than 5 minutes for the whole period 1951 to 2019 when coded in Fortran 95. The algorithm could possibly be accelerated further by taking advantage of the sparsity of the adjacency matrix, since only a few percent of its entries are nonzero.

In this paper, we compared our RCN results with observations over the last 69 years in a year-to-year way, and we could show

that the RCN approach yields useful information on extremes which can complement more conventional methods. Our ultimate goal is to use the RCN method to investigate possible future changes of the frequency and seasonal distribution of extreme events in the future. For climate model projections, one can expect that the years of occurence will vary among the models, so there is no point in year-to-year comparisons. However, our present results let us expect that statistics e.g. over decades can be established reliably. One of our next goals will therefore be to apply RCNs on projections of regional climate models to assess

the future development of extremes and their statistics. From the application perspective it is interesting to use other data sets,

to investigate the impact of spatial resolution and size of the region considered, to apply the RCNs to other regions and to other extremes like floods, and to investigate the relation of the network structure to weather patterns and orography, for example. Also, the incorporation of other relevant information as input like soil moisture and statistics of weather patterns could provide interesting insights.

In this proof-of-concept study we only made use of the most basic properties of networks. Apart from improving the detection of heat waves and other extremes as mentioned above, we also plan to look in more detail at more sophisticated metrics, degree distributions and the appearance and size of communities within the network. From a physics/climatology point of view it is important to understand in more detail why the network measures are able to represent climate dynamics and why their success varies in order to improve the RCN method.

*Code and data availability.*  Codes and data will be made available within the ClimXtreme project

**Appendix A**

*Author contributions.*  GS designed the study, developed computer code and performed the drought part. MB contributed the heat wave part. Both authors shared the preparation of the manuscript.

*Competing interests.*  The authors declare that they have no conflict of interest.

*Acknowledgements.*  This work is funded by the ClimXtreme project within the framework programme "Research for Sustainable Development (FONA3)" by the German Federal Ministry of Education and Research (BMBF) under grants 01LP1902N (G. Schädler and Marcus Breil).

We acknowledge the E-OBS dataset from the EU-FP6 project UERRA and the Copernicus Climate Change Service, and the data providers in the ECA&D project. We also acknowledge support by the KIT-Publication Fund of the Karlsruhe Institute of Technology.

We thank the three reviewers for their comments and suggestions which helped to improve the paper considerably.

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
