# Peer review of "Identification of Droughts and Heat Waves in Germany with Regional Climate Networks"

_Nonlinear Processes in Geophysics, 2020_

## Referee Comment (RC1) · Anonymous Referee #1 · 2 Feb 2021

In this study, the authors used Regional Climate Networks (RCNs) to identify heat waves and droughts in Germany and two sub-regions for the summer half years & summer seasons of the period 1951 to 2019. They used several metrics from RCNs to estimate the extent, intensity and collective behavior of extreme events. The results were compared with standard indices including the effective drought and heat index (EDI and EHI). Their findings suggested that the RCNs are able to identify severe extremes in all cases and moderate extremes in most cases. One highlight of this manuscript is the clear introduction of the concept of RCN. The work is interesting and the RCN methods used in this work may also be useful for other studies. However, there are still several concerns to be addressed. A major revision is needed.

1. One major concern is about the added value of the RCN method compared with the

standard indices (EDI and EHI). In this work, the authors compared the results from the RCN method and those from EDI and EHI, but lack a detailed summary of the advantages of the RCN method.

2. In the manuscript, the authors used different metrics to measure the spatial extent and the intensity of the extreme event. However, detailed discussions on these two properties (extent and intensity) are missing. It seems that the main point of the work is to compare the metrics (mainly the edge density) with the EDI/EHI. It is not clear why the intensity is measured?

3. To determine the correlation threshold, the authors conducted a series of sensitivity runs with respective to correlation threshold and its effect on the metrics. They found that an average edge density of about 0.01 gives good results, and chose a value of 0.95 as correlation threshold. The question is, does this threshold dependent on the distributions of the variable of interest? Can this threshold be used for different variables?

4. When identifying extreme events, a margin of 0.2 is applied to account for averaging and moderate extreme events. Why choose 0.2? Are there any reasons?

5. When calculating the correlations using precipitation time series, dot product was used. However, I think an Event Synchronization (ES) analysis (refer to, e.g., Boers et al., Nature Communications, 2014) may be more appropriate, as in ES, one can consider a time-lag to better determine whether the two considered events from two nodes are synchronized.

6. In table 2 and table 3, the authors compared the metrics in average years and extreme years. Here, a significance test may be needed to better show the differences. In addition, when comparing the different results from average years and extreme years, a composite analysis may be better. For example, calculate the mean metrics over all the average years, and compare them with those averaged over all extreme years.

7. In Fig. 4 and Fig. 9, what do the red and blue curves stand for?

8. On page 19, lines 16-17, the reference Tsonis et al., BAMS, 2006 is repeated twice.
* * *

---

## Referee Comment (RC2) · Anonymous Referee #2 · 2 Feb 2021

With the meteorological data over Germany, the authors investigated the ability of climate network to identify drought and heatwave events. Several network metrics were found to be sensitive to the occurrences of these extreme events. Many droughts and heatwaves correspond to the variations of atmosphere in regional scale rather than local scale. Climate network can inform the spatio-temporal evolution of the regional climate systems, which might be a promising tool for droughts and heatwaves. This work could provide useful reference for studying these extreme events with climate network. However, some necessary information is missing in the manuscript, and some details should be concerned. To better inform the potential readers, I would suggest a major revision. Please find my suggestions in the following.

Major comments:

1. During average years, the distribution of the node degrees is close to the Poisson distribution, characteristic of random networks, while for extreme years the distribution is more uniform and heavy tailed. This is interesting. But the manuscript only demonstrates two cases of 2013 and 2018 for drought in Fig. 4, and two cases of 2002 and 2003 for heatwave in Fig. 9. How about the degree distribution curves in other years? The authors did not mention this in the manuscript. Degree distribution is the essential metric for network approach, please present the degree distribution curves for other years in main text or supplementary text.

2. The choice of the correlation threshold of the time series determines the entries of the adjacency matrix, and this is crucial for the statistical significance and performance of climate network. The authors said that they had conducted the sensitivity runs with respect to correlation threshold. But I would suggest the authors to present more detailed results about your sensitivity runs in the supplementary text.

For example, please use a certain extreme year to illustrate how the choice of correlation threshold influences the shape of the degree distribution curve, and the value of edge density. And please illustrate the similar results for a certain average year.

3. On page 5, lines 1-4, the computation of connection criterion for precipitation data is unusual for climate network links. Please give the citation about this computation method if there is a referred preceding work. Or please give more explanations why use such computation method.

4. On page 16, lines 10, the authors declared that all metrics increase significantly during extreme events, and probability distributions change considerably. However, there is no significant test mentioned in the manuscript. Please present the statistical significance test (such as red noise test) in the main text or supplementary text, to support your conclusions.

5. On page 16, line 3-6, the authors said that 8 of 11 historical summertime heat events can be identified by RCNs. This was identified by the three metrics: edge density, p90

and clustering coefficient. How about the degree distribution? If plotting the degree distribution curves for the 3 missed events, and the 5 false-alarmed events, will they be close to Poisson distribution or uniform distribution? Please give more information and discussion about this.

Minor comments:

1. Please replace the word "resp." with "and" throughout the manuscript, such as on page 1, line 3.

2. On page 3, line 13, the authors defined the dry days as daily precipitation sums less than 1mm/day. Please explain the reason of selecting "1mm/day", or give the citation of preceding works using this definition.

3. On page 4, line 12-14, on. The authors mentioned that, during extreme periods, large scale synchronous behaviour will prevail. For better informing the readers, please give citations that revealed that extreme events can make the synchronous behavior in climate system. The following reference might be helpful.

Faranda, D., Messori, G. Yiou, P. Diagnosing concurrent drivers of weather extremes: application to warm and cold days in North America. Clim Dyn 54, 2187–2201 (2020).

4. In Fig. 1, please give the meaning of the background grey image.

5. In Fig. 2, the image and its curves are blurry, please improve the quality of the image.

6. In Fig. 3, the labels of horizontal axis are unclear. What are the meanings of "edgdens" and "cpp"? They might be edge density and clustering coefficient. These figure labels are not standardized. Please improve them, including the Figs. 6, 7 and 8.

7. In Figs. 4 and 9, please give clear indication of the red and blue curves. Moreover, the label of horizontal axis can be "Network degree", the label of vertical axis can be

"Probability distribution function". Please improve them.

8. In Figs. 3, 6, 7 and 8, please give the significance level for the correlation, such as the p-value.

9. Although most of extreme events can be identified by RCNs in this work, the used Pearson correlation is not the only method to construct climate network. Such as time-lagged Pearson correlation, event synchronization method, mutual information method and causality method, they can be also used to compute the link strength between two network nodes (see the following references). In the summary part, please mention these methods, and please discuss why you selected the Pearson correlation. Or at least mention that these unused methods might be helpful to improve the performance of RCNs on studying heatwaves and droughts.

Donges, J.F., Petrova, I., Loew, A. et al. How complex climate networks complement eigen techniques for the statistical analysis of climatological data. Climate Dynamics, 45, 2407–2424 (2015).

Wang, Y., Gozolchiani, A., Ashkenazy, Y. , Berezin, Y. , Guez, O. , Havlin, S. The dominant imprint of rossby waves in the climate network. Physical Review Letters, 111(13):138501 (2013).

Odenweller, A. , Donner, R. V. Disentangling synchrony from serial dependency in paired-event time series. Physical Review E, 101(5) (2020).

Runge, J. , Petoukhov, V. , Donges, J. F. , Hlinka, J. , Jajcay, N. , Vejmelka, M. , et al. Identifying causal gateways and mediators in complex spatio-temporal systems. Nature Communications, 6, 8502 (2015).

---

## Referee Comment (RC3) · Reik Donner (Referee) · 3 Feb 2021

The authors of this paper present an interesting application of the relatively recent concept of functional climate network analysis to identifying and characterizing droughts and heat waves across Germany. Specifically, they employ a network construction to a subset of the gridded E-OBS data product for daily maximum temperatures and precipitation sums covering entire Germany plus a bit of the surrounding European land area. The presented application is novel in a few aspects: First, the network construction based on the distinction rainy/dry days as opposed to the more often employed selection of "heavy" precipitation days. Second, the focus on time windows of six or three months corresponding to a specific season instead of employing a running window analysis. Third, the consideration of a regional network constructed with a fixed

threshold to the bivariate statistical association measure, instead of a fixation of the edge density that has been more widely employed in recent regional studies. These aspects together provide interesting new insights, yet also call for extended justification and discussion, which I however see only partly provided in the present manuscript. As a consequence, I have a few comments that I would like to invite the authors to consider in revising their work prior to acceptance for publication in NPG.

General comments:

1. There is a vast body on complex network applications in climatology, so it is surprising that the authors essentially cite only some very old papers (Tsonis et al. 2006, Donges et al. 2009) instead of pointing the reader to more thorough recent overviews on the topic (like the review chapter by Donner et al. in the book "Nonlinear and Stochastic Climate Dynamics", 2017; or the book "Networks in Climate" by Dijkstra et al., 2019). In general, referencing the existing literature on climate networks needs to be considerably improved.

2. It is a neat idea to use the association measures between binary yes/no (rain/no rain) sequences describing the precipitation dynamics, yet this obviously throws away all information on precipitation strength with might be valuable in its own. Other works on precipitation based climate networks attempted different strategies – 1) focusing on the timing of locally extreme events only (cf. Malik et al. NPG 2010, Clim. Dyn. 2012; Boers et al. 2013- in a series of papers in GRL, Nature Comm., Clim. Dyn., and most recently Nature; Stolbova et al, NPG 2014, to mention only a few of them, not to request citing those excessively but just to bring the scale of associated publications to the authors' attention), 2) defining an alternative correlation measure replacing the zero precipitation points by the mean rainfall on the rainy days (Ciemer et al., Clim. Dyn., 2018) – I suggest to at least mention those methodological alternatives and motivate more clearly the setting followed in the present work. Regarding the latter, the authors mention the Hamming distance for binary series (p.2, l.14), but appear to use the product of the two sequences (yet ignoring the joint occurrence of no-rain days,

p.5, ll.3-4). This is somewhat confusing and should be clarified from the beginning.

3. As mentioned above, most recent climate network applications have fixed the edge density and let the correlation threshold vary with time, instead of vice versa, the reason being that many network characteristics are directly affected by changing edge densities. In this regard, it is not completely surprising that edge density and clustering coefficient provide rather similar results.

4. Page 3, ll.26-30: I disagree with the authors' interpretation of high values of the clustering coefficient indicating "strong collective behavior" – this is rather represented by a high edge density. The clustering coefficient focuses on transitive connectivity relationships and thereby rather describes the redundancy of connectivity. In a system with spatial autocorrelations implying the linkage probability being distance-dependent, it is likely that a higher edge density implies a slower decay of the spatial autocorrelation and, hence, a higher likelihood of denser (and therefore more transitive/clustered) regional connections. In general, one should keep in mind that the behavior of the clustering coefficient may change with the edge density (and associated with this, the shape/type of the degree distribution). This mutual dependence between the average local clustering coefficient and the degree distribution has been known in network theory for more than 20 years (cf. Barrat and Weigt, EPJB, 2000) and has led to an alternative definition of a global clustering measure often called network transitivity. Moreover, the implications for clustering properties in climate networks at a global scale have been discussed in great detail by Radebach et al., PRE, 2013. The latter paper showed that the behavior of the clustering coefficient under changing conditions can completely reverse if the edge density is varied, while the transitivity provided stable results. Their networks however contained about 10,000 nodes for edge densities of the order of 1% - notably the same order of magnitude as also used in the present work, yet with substantially smaller networks in terms of the number of nodes. It might be interesting (and possibly even relevant for the interpretation of the presented results) to see if similar effects also take place at the regional scale considered in the present work.

Taken together: some of the findings obtained in this study could be well interpreted in the light of the aforementioned references, while there are two questions remaining to me: 1) Why did the authors choose to consider a fixed correlation threshold instead of a fixed edge density? 2) Why did the authors use the mean local clustering coefficient instead of the global network transitivity, which would be less dependent on the edge density and degree distribution?

5. When you consider normalized network metrics accounting for their sample mean and standard deviation, it might look surprising that you only consider the cases >1 and disregard possible cases with normalized values <-1. This might be well justified by the fact that large-scale extreme situations are rather accompanied by elevated values of the network properties (e.g. the elevated transitivity values in global surface air temperature networks along with El Nino/La Nina found by Radebach et al., cited above). However, this aspect should be clarified for readers not familiar with the existing literature on climate network applications.

6. I am strongly concerned about the authors' interpretation of the found degree distributions. For example, on p.4, l.10, they report that they "would expect a network structure resembling a random network". I fundamentally disagree with this statement, especially on such relatively small regional scales. The climate system is always characterized by spatial autocorrelation, implying that the probability of linking two random nodes is not constant (as for a random network in the Erdös-Renyi sense), but depends on the spatial distance. The appropriate null model would therefore be a random geometric graph (e.g. Dall and Christensen, PRE, 2002), which has distinctively different features than an unconstrained random graph. As a result, the authors should carefully revise all statements indicating that a Poissonian degree distribution might provide a good benchmark; in fact, it cannot by construction in the present case.

7. In a similar spirit, the authors report that extreme years see a "more uniform" degree distribution (which I can accept without problems) and have "a heavy tail" (which I cannot accept without the authors showing that the tail is actually heavy, i.e., follows a power-law decay, which I can hardly believe to be the case due to the spatial constraints/boundaries of the constructed regional networks). Please provide corresponding evidence or rephrase.

8. It is interesting that the edge density for the whole network is commonly smaller than in the two subnetworks. Does this imply a reduced presence of North-South links connecting the northern and southern subdomains? I think it might be interesting to study this further, e.g., by using directional network properties like in Rheinwalt et al., Clim. Dyn., 2015, or Wolf et al., PRE, 2019. (But this is more a suggestion for follow-up works.)

9. Still related to the results summarized in Tab. 2: Can you add information on the spread among the extreme/normal years instead of showing only two examples that might reflect either an unbiased or a biased selection? Just the numbers as they stand now do hardly present relevant information in a proper statistical sense.

10. I recommend some improvements on the artwork (larger axis labels/ticks with proper symbols/words instead of cryptic abbreviations as labels). The figure captions should be self-explanatory (e.g., I do not find information about the meanings of the red lines in Fig. 9).

11. When comparing the network classifications with the EDI/EHI based classifications, I recommend to avoiding using the terminology of "false" classifications (maybe rather use "inconsistent"), since neither of the methods presents something that should be considered a "ground truth". Even EDI/EHI are not necessarily the "gold standard" to be used for benchmarking drought/heat wave years/seasons.

Minor suggestions:

- P.1, ll.7-9: "Metrics to identify extremes. . ." sounds a bit bold, I'd rather suggest some more explicit formulation like "Metrics to identify extraordinary network configurations. . . during years with extraordinary drought or heat conditions."

- P.2, ll.8-9: Is it really necessary that extreme events have such large spatial scales? I believe this depends on the type of extreme and its associated temporal scale – while being correct for seasonal-scale extremes discussed in this work, it might be questionable for some short-lived extremes like heavy rainfall due to almost stationary convective (thunderstorm) activity.

- P.2, l.13: Instead of "correlation measure", I suggest using the more general term "statistical association" measure. The Hamming distance is not a correlation measure in the usual statistical sense.

- P.4, l.2: "limit of a binomial distribution" should be clarified a bit, you probably refer to the fact that the probability distribution for each single link to exist is binomial.

- P.4, l.5: Mutual information is NOT(!) a network measure like the other mentioned quantities, but a different statistical association measure for constructing the networks.

- P.9, ll.21-22: Does this mean that you consider this finding mainly an effect of temporal rather than spatial autocorrelation?

- P.9, l.23: I would rather interpret lower clustering coefficients as indicative of (spatially) more "fragmented" connectivity structures, which would match with the statements regarding orography and land use.

- P.11, ll.4-5: I am somewhat surprised about this statement. To my best knowledge, the northernmost and especially northwestern part of Germany has a climate that I would hardly characterize as continental.

In summary, while the paper is interesting and well written (a few typos need to be corrected), I feel that the work is not sufficiently mature in the sense of being properly presented and interpreted in the context of the existing topically relevant literature. I therefore recommend the authors to perform a major revision along the lines of my above comments before further consideration of this work for final publication in NPG.

---

## Author Comment (AC1) · 2 Mar 2021

In this study, the authors used Regional Climate Networks (RCNs) to identify heat
waves and droughts in Germany and two sub-regions for the summer half years &
summer seasons of the period 1951 to 2019. They used several metrics from RCNs
to estimate the extent, intensity and collective behavior of extreme events. The results
were compared with standard indices including the effective drought and heat index
(EDI and EHI). Their findings suggested that the RCNs are able to identify severe
extremes in all cases and moderate extremes in most cases. One highlight of this
manuscript is the clear introduction of the concept of RCN. The work is interesting and
the RCN methods used in this work may also be useful for other studies. However,
there are still several concerns to be addressed. A major revision is needed.

Dear referee,
thank you for your time and your constructive comments.
In response to the suggestions and comments of the three referees, we have made major
changes in the manuscript in addition to specific answers to your comments:
- we added a section „Sensitivity of the metrics to correlation thresholds"
- we rewrote the „Comparison of the RCN results with other extreme indices" section.
  The figures are replaced by tables comparing the RCN metrics for a range of
  correlation thresholds with EDI/EHI.
- We focus now on the edge density as the relevant metric
Please find our answers to your specific comments on the previous manuscript below (in
blue).

1. One major concern is about the added value of the RCN method compared with the
standard indices (EDI and EHI). In this work, the authors compared the results from
the RCN method and those from EDI and EHI, but lack a detailed summary of the
advantages of the RCN method.
- It is an alternative, easy-to-apply and computationally efficient method which can be
  integrated into the post-processing of climate model output to provide statistics of
  extreme events (change of frequency, seasons of occurence, …)
- it complements standard methods to detect extremes
- it allows to study whole regions and seasons
We have expanded a bit more on this in the introduction and the summary.

2. In the manuscript, the authors used different metrics to measure the spatial extent
and the intensity of the extreme event. However, detailed discussions on these two
properties (extent and intensity) are missing. It seems that the main point of the work
is to compare the metrics (mainly the edge density) with the EDI/EHI. It is not clear why
the intensity is measured?
You are right, the main point of the work is indeed to compare the edge density with the
EDI/EHI, and we do not consider intensity, which is omitted now.

3. To determine the correlation threshold, the authors conducted a series of sensitivity

runs with respective to correlation threshold and its effect on the metrics. They found that an average edge density of about 0.01 gives good results, and chose a value of 0.95 as correlation threshold. The question is, does this threshold dependent on the distributions of the variable of interest? Can this threshold be used for different variables?

We discuss this in the sensitivity section. In the modified paper, we use now the same three threshold values (0.85,0.90 and 0.95) for heat waves and droughts for the comparisons with EDI and EHI. As detailed in the sensitivity section, the results do not differ much for these threshold values, as long as they are around 0.9. However, the results for droughts are better than the ones for heat waves, which might have more to do with the construction of the network than with the choice of the thresholds.

4. When identifying extreme events, a margin of 0.2 is applied to account for averaging and moderate extreme events. Why choose 0.2? Are there any reasons?

This was confusing indeed. We omitted it and just look at values > 1 (or < -1 for EDI), but mention „border cases" with values just around 1. Note however that this entails changes in the detection of extremes.

5. When calculating the correlations using precipitation time series, dot product was used. However, I think an Event Synchronization (ES) analysis (refer to, e.g., Boers et al., Nature Communications, 2014) may be more appropriate, as in ES, one can consider a time-lag to better determine whether the two considered events from two nodes are synchronized.

It seems to us that ES is more suitable for short (e.g. daily) time scales and instationary situations. Since we are looking here at much longer (seasonal) and quasi-stationary time scales, we do not think that ES will offer advantages. But we will experiment with ES in upcoming studies. Especially in the context of heat waves, this might improve our results.

6. In table 2 and table 3, the authors compared the metrics in average years and extreme years. Here, a significance test may be needed to better show the differences. In addition, when comparing the different results from average years and extreme years, a composite analysis may be better. For example, calculate the mean metrics over all the average years, and compare them with those averaged over all extreme years.

We followed your suggestion in the sensitivity section and performed a Wilcoxon significance test.

7. In Fig. 4 and Fig. 9, what do the red and blue curves stand for?

These figures do not appear any more. Blue was the RCN degree distribution, red the corresponding Poisson distribution.

8. On page 19, lines 16-17, the reference Tsonis et al., BAMS, 2006 is repeated twice.

Corrected.

---

## Author Comment (AC2) · 2 Mar 2021

With the meteorological data over Germany, the authors investigated the ability of climate
network to identify drought and heatwave events. Several network metrics were
found to be sensitive to the occurrences of these extreme events. Many droughts and
heatwaves correspond to the variations of atmosphere in regional scale rather than local
scale. Climate network can inform the spatio-temporal evolution of the regional climate
systems, which might be a promising tool for droughts and heatwaves. This work
could provide useful reference for studying these extreme events with climate network.
However, some necessary information is missing in the manuscript, and some details
should be concerned. To better inform the potential readers, I would suggest a major
revision. Please find my suggestions in the following.

Dear referee,
thank you for your time and your constructive comments.
In response to the suggestions and comments of the three referees, we have made major
changes in the manuscript in addition to specific answers to your comments:
- we added a section „Sensitivity of the metrics to correlation thresholds"
- we rewrote the „Comparison of the RCN results with other extreme indices" section.
  The figures are replaced by tables comparing the RCN metrics for a range of
  correlation thresholds with EDI/EHI.
- We focus now on the edge density as the relevant metric
Please find our answers to your specific comments below (in blue).

Major comments:

1. During average years, the distribution of the node degrees is close to the Poisson
distribution, characteristic of random networks, while for extreme years the distribution
is more uniform and heavy tailed. This is interesting. But the manuscript only demonstrates
two cases of 2013 and 2018 for drought in Fig. 4, and two cases of 2002 and
2003 for heatwave in Fig. 9. How about the degree distribution curves in other years?
The authors did not mention this in the manuscript. Degree distribution is the essential
metric for network approach, please present the degree distribution curves for other
years in main text or supplementary text.
We show and discuss cumulative degree distributions for normal and extreme years in the
sensitivity section. We must admit that there was some wishful thinking along „normal ~
Poisson, extreme ~ flat and heavy tailed" on our side which we could not substantiate. We
therefore omitted these considerations, nice as they would have been.

2. The choice of the correlation threshold of the time series determines the entries of
the adjacency matrix, and this is crucial for the statistical significance and performance
of climate network. The authors said that they had conducted the sensitivity runs with
respect to correlation threshold. But I would suggest the authors to present more
detailed results about your sensitivity runs in the supplementary text.

For example, please use a certain extreme year to illustrate how the choice of correlation threshold influences the shape of the degree distribution curve, and the value of edge density. And please illustrate the similar results for a certain average year.

We did this in the new sensitivity section

3. On page 5, lines 1-4, the computation of connection criterion for precipitation data is unusual for climate network links. Please give the citation about this computation method if there is a referred preceding work. Or please give more explanations why use such computation method.

Our formulation was a bit unclear and long-winded there, the correlation coefficient was used. We rephrased the sentences.

4. On page 16, lines 10, the authors declared that all metrics increase significantly during extreme events, and probability distributions change considerably. However, there is no significant test mentioned in the manuscript. Please present the statistical significance test (such as red noise test) in the main text or supplementary text, to support your conclusions.

We expanded on this and included a Wilcoxon test normal vs. extreme years in the sensitivity section.

5. On page 16, line 3-6, the authors said that 8 of 11 historical summertime heat events can be identified by RCNs. This was identified by the three metrics: edge density, p90 and clustering coefficient. How about the degree distribution? If plotting the degree distribution curves for the 3 missed events, and the 5 false-alarmed events, will they be close to Poisson distribution or uniform distribution? Please give more information and discussion about this.

We compare degree distributions in the sensitivity section. As we said in our response to your point 1 above, we could not substantiate the Poisson argument, and the attribution of distributions is unfortunately not as clear cut as we had assumed (see section on sensitivity in the revised paper).

Minor comments:

1. Please replace the word "resp." with "and" throughout the manuscript, such as on page 1, line 3.

Done

2. On page 3, line 13, the authors defined the dry days as daily precipitation sums less than 1mm/day. Please explain the reason of selecting "1mm/day", or give the citation of preceding works using this definition.

We use the E-Obs-definition of dry days described in: EUMETNET/ECSN optional programme: 'European Climate Assessment & Dataset (ECA&D)'
Algorithm Theoretical Basis Document (ATBD), or see
https://www.ecad.eu/FAQ/index.php#5

3. On page 4, line 12-14, on. The authors mentioned that, during extreme periods, large scale synchronous behaviour will prevail. For better informing the readers, please give citations that revealed that extreme events can make the synchronous behavior in climate system. The following reference might be helpful.

We are not shure if we understand your comment. In our context, extremes are characterised by the appearance of high temperatures or no precipitation in a region over an extended period of time. This is what we mean by „synchronous behaviour".

4. In Fig. 1, please give the meaning of the background grey image.
Done (relief of Germany)
5. In Fig. 2, the image and its curves are blurry, please improve the quality of the image.
We omitted this figure in the revised manuscript.
6. In Fig. 3, the labels of horizontal axis are unclear. What are the meanings of "edgdens" and "cpp"? They might be edge density and clustering coefficient. These figure labels are not standardized. Please improve them, including the Figs. 6, 7 and 8.
These figures are replaced by tables in the revised manuscript.
7. In Figs. 4 and 9, please give clear indication of the red and blue curves. Moreover, the label of horizontal axis can be "Network degree", the label of vertical axis can be "Probability distribution function". Please improve them.
These figures do not appear any more in the revised manuscript. Blue was the RCN degree distribution, red the corresponding Poisson distribution.
8. In Figs. 3, 6, 7 and 8, please give the significance level for the correlation, such as the p-value.
In the revised manuscript, we provide tables for comparison.
9. Although most of extreme events can be identified by RCNs in this work, the used Pearson correlation is not the only method to construct climate network. Such as timelagged Pearson correlation, event synchronization method, mutual information method and causality method, they can be also used to compute the link strength between two network nodes (see the following references). In the summary part, please mention these methods, and please discuss why you selected the Pearson correlation. Or at least mention that these unused methods might be helpful to improve the performance of RCNs on studying heatwaves and droughts.
Donges, J.F., Petrova, I., Loew, A. et al. How complex climate networks complement eigen techniques for the statistical analysis of climatological data. Climate Dynamics, 45, 2407–2424 (2015).
Wang, Y., Gozolchiani, A., Ashkenazy, Y. , Berezin, Y. , Guez, O. , Havlin, S. The dominant imprint of rossby waves in the climate network. Physical Review Letters, 111(13):138501 (2013).
Odenweller, A. , Donner, R. V. Disentangling synchrony from serial dependency in paired-event time series. Physical Review E, 101(5) (2020).
Runge, J. , Petoukhov, V. , Donges, J. F. , Hlinka, J. , Jajcay, N. , Vejmelka, M. , et al. Identifying causal gateways and mediators in complex spatio-temporal systems. Nature Communications, 6, 8502 (2015).
Thank you for drawing our attention to these papers. We mentioned these methods in sec 2.1 and the summary of the revised manuscript and will use some of them in an upcoming study.

---

## Author Comment (AC3) · 2 Mar 2021

Reik Donner (Referee) reik.donner@h2.de

The authors of this paper present an interesting application of the relatively recent concept of functional climate network analysis to identifying and characterizing droughts and heat waves across Germany. Specifically, they employ a network construction to a subset of the gridded E-OBS data product for daily maximum temperatures and precipitation sums covering entire Germany plus a bit of the surrounding European land area. The presented application is novel in a few aspects: First, the network construction based on the distinction rainy/dry days as opposed to the more often employed selection of "heavy" precipitation days. Second, the focus on time windows of six or three months corresponding to a specific season instead of employing a running window analysis. Third, the consideration of a regional network constructed with a fixed threshold to the bivariate statistical association measure, instead of a fixation of the edge density that has been more widely employed in recent regional studies. These aspects together provide interesting new insights, yet also call for extended justification and discussion, which I however see only partly provided in the present manuscript. As a consequence, I have a few comments that I would like to invite the authors to consider in revising their work prior to acceptance for publication in NPG.

Dear Prof. Donner,
thank you for your time and your constructive comments.
In response to the suggestions and comments of the three referees, we have made major changes in the manuscript in addition to specific answers to your comments:
- we added a section „Sensitivity of the metrics to correlation thresholds"
- we rewrote the „Comparison of the RCN results with other extreme indices" section. The figures are replaced by tables comparing the RCN metrics for a range of correlation thresholds with EDI/EHI.
- We focus now on the edge density as the relevant metric
Please find our answers to your specific comments below (in blue).

General comments:
1. There is a vast body on complex network applications in climatology, so it is surprising that the authors essentially cite only some very old papers (Tsonis et al. 2006, Donges et al. 2009) instead of pointing the reader to more thorough recent overviews on the topic (like the review chapter by Donner et al. in the book "Nonlinear and Stochastic Climate Dynamics", 2017; or the book "Networks in Climate" by Dijkstra et al., 2019). In general, referencing the existing literature on climate networks needs to be considerably improved
Thank you for pointing us to the references. We have added recent publications relevant for our topic.

2. It is a neat idea to use the association measures between binary yes/no (rain/norain) sequences describing the precipitation dynamics, yet this obviously throws away all information on precipitation strength with might be valuable in its

own. Other works on precipitation based climate networks attempted different strategies –

An in between remark: since we are interested in droughts, we only have the 0-1 decision wet-dry days, so precipitation strength does not concern us, apart from the definition dry day = day with precipitation sum < 1 mm.

1) focusing on the timing of locally extreme events only (cf. Malik et al. NPG 2010, Clim. Dyn.2012; Boers et al. 2013- in a series of papers in GRL, Nature Comm., Clim. Dyn., and most recently Nature; Stolbova et al, NPG 2014, to mention only a few of them, not to request citing those excessively but just to bring the scale of associated publications to the authors' attention),

Thank you for drawing our attention to these papers.

2) defining an alternative correlation measure replacing the zero precipitation points by the mean rainfall on the rainy days (Ciemer et al., Clim.Dyn., 2018) – I suggest to at least mention those methodological alternatives and motivate more clearly the setting followed in the present work. Regarding the latter, the authors mention the Hamming distance for binary series (p.2, l.14), but appear to use the product of the two sequences (yet ignoring the joint occurrence of no-rain days, p.5, ll.3-4). This is somewhat confusing and should be clarified from the beginning.

Our formulation was a bit unclear and long-winded there – we rephrased the sentences.

3. As mentioned above, most recent climate network applications have fixed the edge density and let the correlation threshold vary with time, instead of vice versa, the reason being that many network characteristics are directly affected by changing edge densities. In this regard, it is not completely surprising that edge density and clustering coefficient provide rather similar results.

We agree and do not use the clustering coefficient for the comparisons in the revised manuscript. Some discussion is provided in the new sensitivity section.

4. Page 3, ll.26-30: I disagree with the authors' interpretation of high values of the clustering coefficient indicating "strong collective behavior" – this is rather represented by a high edge density. The clustering coefficient focuses on transitive connectivity relationships and thereby rather describes the redundancy of connectivity. In a system with spatial autocorrelations implying the linkage probability being distance-dependent,it is likely that a higher edge density implies a slower decay of the spatial autocorrelation and, hence, a higher likelihood of denser (and therefore more transitive/clustered regional connections. In general, one should keep in mind that the behavior of the clustering coefficient may change with the edge density (and associated with this, theshape/type of the degree distribution). This mutual dependence between the average local clustering coefficient and the degree distribution has been known in network theory for more than 20 years (cf. Barrat and Weigt, EPJB, 2000) and has led to an alternative definition of a global clustering measure often called network transitivity. Moreover, the implications for clustering properties in climate networks at a global scale have been discussed in great detail by Radebach et al., PRE, 2013. The latter paper showed that the behavior of the clustering coefficient under changing conditions can completely reverse if the edge density is varied, while the transitivity provided stable results. Their networks however contained about 10,000 nodes for edge densities of the order of 1% - notably the same order of magnitude as also used in the present work, yet with

substantially smaller networks in terms of the number of nodes. It might be interesting (and possibly even relevant for the interpretation of the presented results) to see if similar effects also take place at the regional scale considered in the present work.

Taken together: some of the findings obtained in this study could be well interpreted in the light of the aforementioned references, while there are two questions remaining to me:

1) Why did the authors choose to consider a fixed correlation threshold instead of a fixed edge density?

As we explained in the introduction, we associate regional extremes with pronounced collective behavior, i.e. high edge density for a given correlation threshold. The primary climatically given is the time series correlation, and the network edge density reflects the correlation pattern. For us the year-to-year changes of the edge density and not the edge density itself is the interesting quantity to detect extremes. There is also the problem that exact fixing of the edge density is not possible and the question as to which edge density to choose – values in the literature spread over quite a range. We discuss this to some extent in the sensitivity section.

2) Why did the authors use the mean local clustering coefficient instead of the global network transitivity, which would be less dependent on the edge density and degree distribution?

Since edge density and clustering coefficient are highly correlated, we only consider the edge density for the comparisons in the new manuscript. We were not aware of the advantages of the network transitivity – thank you for the hint!

5. When you consider normalized network metrics accounting for their sample mean and standard deviation, it might look surprising that you only consider the cases >1and disregard possible cases with normalized values <-1. This might be well justified by the fact that large-scale extreme situations are rather accompanied by elevated values of the network properties (e.g. the elevated transitivity values in global surface air temperature networks along with El Nino/La Nina found by Radebach et al., cited above). However, this aspect should be clarified for readers not familiar with the existing literature on climate network applications.

Negative values of the normalized edge density would mean edge densities are below average, i.e. nodes are less correlated. This could be due to a) the times series contains mostly zeroes, i.e. the year has less-than-average dry days, i.e. it is a „wet" year, or b) the time series of the nodes are quite uncorrelated, i.e. we have mainly small-scale events. Both possibilites would not be indicative of the kind of extremes we are looking for. Probably possibility a) is prevailing, since in some cases we looked at, we have values < -1 combined with EDI > 1, i.e a wet year.

6. I am strongly concerned about the authors' interpretation of the found degree distributions. For example, on p.4, l.10, they report that they "would expect a network structure resembling a random network". I fundamentally disagree with this statement, especially on such relatively small regional scales. The climate system is always characterized by spatial autocorrelation, implying that the probability of linking two random nodes is not constant (as for a random network in the Erdös-Renyi sense), but depends on the spatial distance. The appropriate null model would therefore be a random geometric graph (e.g. Dall and Christensen, PRE, 2002), which has distinctively different features than an unconstrained random graph. As a

result, the authors should carefully revise all statements indicating that a Poissonian degree distribution might provide a good benchmark; in fact, it cannot by construction in the present case.

As we said in the reply to referee #2, there was some wishful thinking from our side involved. We could not substantiate „normal ~ Poisson, extreme ~ flat and heavy tailed" and therefore omitted these considerations.

7. In a similar spirit, the authors report that extreme years see a "more uniform" degree distribution (which I can accept without problems) and have "a heavy tail" (which I cannot accept without the authors showing that the tail is actually heavy, i.e., follows a power-law decay, which I can hardly believe to be the case due to the spatialconstraints/boundaries of the constructed regional networks). Please provide corre-sponding evidence or rephrase.

We plotted the (logarithmic) cumulative degree distribution for several average and extreme years. For degrees higher than about 0.7*maximum degree, in most cases a straight line appeared (see exemplary figures below: they show the log of the cumulative distribution plotted over log of the node degrees for 1976: extreme drought (right) and 2013: average year (left), but the exponent (around 6) is much higher than the expected values (about 2 to 3) , which makes a power law behaviour at least doubtful (especially since there are many more possible distribution candidates). We therefore rephrased our statement.

[Figure]

8. It is interesting that the edge density for the whole network is commonly smaller than in the two subnetworks. Does this imply a reduced presence of North-South links connecting the northern and southern subdomains? I think it might be interesting to study this further, e.g., by using directional network properties like in Rheinwalt et al.,Clim. Dyn., 2015, or Wolf et al., PRE, 2019. (But this is more a suggestion for follow-upworks.)

We did not look into this – it is certainly worth further investigation. It could also be an indication of the existence of separate „northern" and „southern" communities.

9. Still related to the results summarized in Tab. 2: Can you add information on the spread among the extreme/normal years instead of showing only two examples that might reflect either an unbiased or a biased selection? Just the numbers as they stand now do hardly present relevant information in a proper statistical sense.

A discussion of the ratios extreme/normal years can now be found in the new sensitivity section.

10. I recommend some improvements on the artwork (larger axis labels/ticks withproper symbols/words instead of cryptic abbreviations as labels). The figure captionsshould be self-explanatory (e.g., I do not find information about the meanings of the red lines in Fig. 9).
Done. In the (now omitted) Fig. 9 blue was the RCN degree distribution, red the corresponding Poisson distribution.
11. When comparing the network classifications with the EDI/EHI based classifications,I recommend to avoiding using the terminology of "false" classifications (maybe rather use "inconsistent"), since neither of the methods presents something that should be considered a "ground truth". Even EDI/EHI are not necessarily the "gold standard" to be used for benchmarking drought/heat wave years/seasons.
Done.

Minor suggestions:-
-P.1, ll.7-9: "Metrics to identify extremes..." sounds a bit bold, I'd rather suggest somemore explicit formulation like "Metrics to identify extraordinary network configurations...during years with extraordinary drought or heat conditions."
Changed.
-P.2, ll.8-9: Is it really necessary that extreme events have such large spatial scales?I believe this depends on the type of extreme and its associated temporal scale –while being correct for seasonal-scale extremes discussed in this work, it might be questionable for some short-lived extremes like heavy rainfall due to almost stationary convective (thunderstorm) activity.
You are right, what is considered an extreme depends on type and temporal scale. In our paper, we are not looking for small-scale or short lived events. The type of extremes we are looking for is the one affecting whole regions (in the sense described, e.g. geographical regions, large conurbations, catchment areas, ...) over an extended time span (e.g. a season).
-P.2, l.13: Instead of "correlation measure", I suggest using the more general term"statistical association" measure. The Hamming distance is not a correlation measure in the usual statistical sense.
Done.
- P.4, l.2: "limit of a binomial distribution" should be clarified a bit, you probably refer to the fact that the probability distribution for each single link to exist is binomial.
We omitted now the Poisson part.
- P.4, l.5: Mutual information is NOT(!) a network measure like the other mentioned quantities, but a different statistical association measure for constructing the networks.
Corrected.
-P.9, ll.21-22: Does this mean that you consider this finding mainly an effect of temporal rather than spatial autocorrelation?
No. It just seems that the signal/noise level is lower during normal years.
- P.9, l.23: I would rather interpret lower clustering coefficients as indicative of (spatially) more "fragmented" connectivity structures, which would match with the statements regarding orography and land use.
Changed.

- P.11, ll.4-5: I am somewhat surprised about this statement. To my best knowledge,the northernmost and especially northwestern part of Germany has a climate that I would hardly characterize as continental.

We meant the eastern parts – corrected.

In summary, while the paper is interesting and well written (a few typos need to be corrected), I feel that the work is not sufficiently mature in the sense of being properly presented and interpreted in the context of the existing topically relevant literature. I therefore recommend the authors to perform a major revision along the lines of my above comments before further consideration of this work for final publication in NPG.C6

---

## Author Response (AR2)

Editor Decision on "Identification of Droughts
and Heat Waves in Germany with Regional Climate
Networks" by Gerd Schädler and Marcus Breil
from 23 Mar 2021

Dear Christian Franzke,

thank you for accepting our paper. Please find our answers to the second round of reviewer comments below (in blue).

Reviewer #1

The authors have addressed all my concerns and the manuscript has been improved considerably. I would like to recommend accepting this manuscript, after the following minor point is addressed.
On page 6, lines 17-18, using EDI, droughts are defined "when the spatially and temporally averaged EDI is less than -1". However, on page 16, lines 14-15, "In the years 1973 and 1996", extreme years in EDI, EDI is just above the threshold (value 1.03)". I am a bit confused by the word "above", and I guess the EDI value should be negative, right?
Thank you for spotting this mistake. Since we are looking at negative EDI values, it should read „below" and „value -1.03".
In addition, on page 17, line 1, "an additional extreme year in EDI just below the treshold" (here, a typo, "treshold" should be "threshold"), here you use the word "below". I guess there might be some typos?
Corrected.

Reviewer #2

The authors have addressed most of my concerns. In the revised manuscript, the authors showed the sensitivity tests about their results, and I agree with them since that the test of statistical significance is important to the usage of climate network approach. Generally, this work demonstrates that climate network is able to be an effective tool in distinguishing some of drought and heatwave events, and it could be a useful reference for the future work about heatwaves and climate network. I would be oriented to suggest accepting the paper as it is for publication on NPG.
However, I have some considerations that authors and editors may take into account if they agree and think are useful to improve the manuscript:
(1) In the previous version of the manuscript, the authors showed the node degree probability distributions for extreme years and normal years (in the Fig. 4 of the previous version): During average years, the distribution of the node degrees is close to the Poisson distribution, characteristic of random networks, while for extreme years the distribution is more uniform and heavy tailed. It is a fact for the existence of such phenomenon observed from the authors' results. This is interesting and impressive.
However, in the revised manuscript, this figure was omitted, since the authors said they could not substantiate or explain the underlying mechanisms. I would suggest showing this figure in the revised manuscript, or in the supplementary materials. The authors may only give brief introductions for the figure. Though this phenomenon has not been thoroughly explained at the current stage, at least its existence should be informed to the readers. This could be useful for the future work which refers this manuscript.

We have (re-)included a brief paragraph on this observation. It is certainly worthwhile to investigate this further in the context of random geometric graphs (which can be quite similar to random graphs with Poisson distributions under certain conditions).

(2) A recent work addressed the heatwave patterns and propagations using climate network (see below), and it may be mentioned in the manuscript. This could be useful for better informing the readers.

Mondal S, Mishra AK. Complex networks reveal heatwave patterns and propagations over the USA. Geophysical Research Letters, 48 (2021).

We mention the paper in the introduction.

(3) I would suggest the authors to revise the captions of Tabs. 3 and 4, and Figs. 2, 5, 6, 7, 8 and 9. The information in the captions was omitted too much, such that it was not convenient for readers.

We hope the captions are clearer now.